# Building the Bridge of Schrödinger:
# A Continuous Entropic Optimal Transport Benchmark

**Nikita Gushchin**
Skoltech*
Moscow, Russia
n.gushchin@skoltech.ru

**Alexander Kolesov**
Skoltech*
Moscow, Russia
a.kolesov@skoltech.ru

**Petr Mokrov**
Skoltech*
Moscow, Russia
petr.mokrov@skoltech.ru

**Polina Karpikova**
Skoltech*
Moscow, Russia
polina.karpikova@skoltech.ru

**Andrey Spiridonov**
Skoltech*
Moscow, Russia
andrew.spiridonov@skoltech.ru

**Evgeny Burnaev**
Skoltech*
AIRI†
Moscow, Russia
e.burnaev@skoltech.ru

**Alexander Korotin**
Skoltech*
AIRI†
Moscow, Russia
a.korotin@skoltech.ru

## Abstract

Over the last several years, there has been significant progress in developing neural solvers for the Schrödinger Bridge (SB) problem and applying them to generative modelling. This new research field is justifiably fruitful as it is interconnected with the practically well-performing diffusion models and theoretically grounded entropic optimal transport (EOT). Still, the area lacks non-trivial tests allowing a researcher to understand how well the methods solve SB or its equivalent continuous EOT problem. We fill this gap and propose a novel way to create pairs of probability distributions for which the ground truth OT solution is known by the construction. Our methodology is generic and works for a wide range of OT formulations, in particular, it covers the EOT which is equivalent to SB (the main interest of our study). This development allows us to create continuous benchmark distributions with the known EOT and SB solutions on high-dimensional spaces such as spaces of images. As an illustration, we use these benchmark pairs to test how well existing neural EOT/SB solvers actually compute the EOT solution. Our code for constructing benchmark pairs under different setups is available at:
https://github.com/ngushchin/EntropicOTBenchmark
.

Diffusion models are a powerful tool to solve image synthesis [25, 45] and image-to-image translation [50, 47] tasks. Still, they suffer from the time-consuming inference which requires modeling thousands of diffusion steps. Recently, the **Schrodinger Bridge** (SB) has arisen as a promising framework to cope with this issue [15, 9, 54]. Informally, SB is a special diffusion which has rather *straight trajectories* and *finite time horizon*. Thus, it may require fewer discretization steps to infer the diffusion.

---

*Skolkovo Institute of Science and Technology
†Artificial Intelligence Research Institute

37th Conference on Neural Information Processing Systems (NeurIPS 2023) Track on Datasets and Benchmarks.

In addition to promising practical features, SB is known to have good and well-studied theoretical properties. Namely, it is equivalent [38] to the **Entropic Optimal Transport** problem (EOT, [13, 20]) about moving the mass of one probability distribution to the other in the most efficient way. This problem has gained a genuine interest in the machine learning community thanks to its nice sample complexity properties, convenient dual form and a wide range of applications [28, 6, 44].

Expectedly, recent **neural EOT/SB** solvers start showing promising performance in various tasks [15, 52, 9, 14, 23, 42]. However, it remains unclear to which extent this success is actually attributed to the fact that these methods properly solve EOT/SB problem rather than to a good choice of parameterization, regularization, tricks, etc. This ambiguity exists because of the **lack of ways to evaluate the performance of solvers qualitatively** in solving EOT/SB. Specifically, the class of continuous distributions with the analytically known EOT/SB solution is narrow (Gaussians [10, 41, 26, 7]) and these solutions have been obtained only recently. Hence, although papers in the field of neural EOT/SB frequently appear, we never know how well they actually solve EOT/SB.

**Contributions**. We develop a generic methodology for evaluating continuous EOT/SB solvers.

1. We propose a generic method to create continuous pairs of probability distributions with analytically known (by our construction) EOT solution between them (§3.1, §3.2).

2. We use log-sum-exp of quadratic functions (§3.3, §3.4) to construct pairs of distributions (§4) that we use as a benchmark with analytically-known EOT/SB solution for the quadratic cost.

3. We use these **benchmark pairs** to evaluate (§5) many popular neural EOT/SB solvers (§2) in high-dimensional spaces, including the space of $64 \times 64$ celebrity faces.

In the field of neural OT, there already exist several benchmarks for the Wasserstein-2 [32], the Wasserstein-1 [31] and the Wasserstein-2 barycenter [31] OT tasks. Their benchmark construction methodologies work **only** for specific OT formulations and **do not** generalize to EOT which we study.

# 1 Background: Optimal Transport and Schrödinger Bridges Theory

We work in Euclidean space $\mathcal{X} = \mathcal{Y} = \mathbb{R}^D$ equipped with the standard Euclidean norm $\|\cdot\|$. We use $\mathcal{P}(\mathcal{X}) = \mathcal{P}(\mathcal{Y}) = \mathcal{P}(\mathbb{R}^D)$ to denote the sets of Borel probability distributions on $\mathcal{X}, \mathcal{Y}$, respectively.

**Classic (Kantorovich) OT formulation** [27, 53, 48]. For two distributions $\mathbb{P}_0 \in \mathcal{P}(\mathcal{X})$, $\mathbb{P}_1 \in \mathcal{P}(\mathcal{Y})$ and a cost function $c : \mathcal{X} \times \mathcal{Y} \to \mathbb{R}$, consider the following problem (Fig. 1a):

$$\text{OT}_c(\mathbb{P}_0, \mathbb{P}_1) \stackrel{\text{def}}{=} \inf_{\pi \in \Pi(\mathbb{P}_0, \mathbb{P}_1)} \int_{\mathcal{X} \times \mathcal{Y}} c(x, y) d\pi(x, y), \tag{1}$$

where the optimization is performed over the set $\Pi(\mathbb{P}_0, \mathbb{P}_1)$ of transport plans, i.e., joint distributions on $\mathcal{X} \times \mathcal{Y}$ with marginals $\mathbb{P}_0, \mathbb{P}_1$, respectively. The set $\Pi(\mathbb{P}_0, \mathbb{P}_1)$ is non-empty as it always contains the trivial plan $\mathbb{P}_0 \times \mathbb{P}_1$. With mild assumptions, a minimizer $\pi^*$ of (1) exists and is called an *OT plan*. Typical examples of $c$ are powers of Euclidean norms, i.e., $c(x, y) = \frac{1}{q}\|x - y\|^q$, $q \geq 1$.

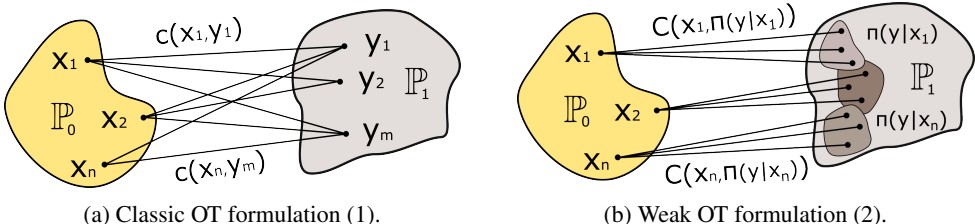

(a) Classic OT formulation (1).        (b) Weak OT formulation (2).

Figure 1: Classic (Kantorovich's) and weak OT formulations.

**Weak OT formulation** [22, 2, 3]. Let $C : \mathcal{X} \times \mathcal{P}(\mathcal{Y}) \to \mathbb{R} \cup \{+\infty\}$ be a weak cost which takes a point $x \in \mathcal{X}$ and a distribution of $y \in \mathcal{Y}$ as inputs. The weak OT cost between $\mathbb{P}_0, \mathbb{P}_1$ is (Fig. 1b)

$$\text{WOT}_C(\mathbb{P}_0, \mathbb{P}_1) \stackrel{\text{def}}{=} \inf_{\pi \in \Pi(\mathbb{P}_0, \mathbb{P}_1)} \int_{\mathcal{X}} C(x, \pi(\cdot|x)) d\pi_0(x) = \inf_{\pi \in \Pi(\mathbb{P}_0, \mathbb{P}_1)} \int_{\mathcal{X}} C(x, \pi(\cdot|x)) d\mathbb{P}_0(x), \tag{2}$$

where $\pi(\cdot|x)$ denotes the conditional distribution of $y \in \mathcal{Y}$ given $x \in \mathcal{X}$ and $\pi_0$ is the projection of $\pi$ to $\mathcal{X}$ which equals $\mathbb{P}_0$ since $\pi \in \Pi(\mathbb{P}_0, \mathbb{P}_1)$. Weak OT formulation (2) generalizes classic OT (1): it suffices to pick $C(x, \pi(\cdot|x)) = \int_{\mathcal{Y}} c(x, y) d\pi(y|x)$ to obtain (1) from (2). A more general case of a

weak cost is $C(x, \pi(\cdot|x)) = \int_{\mathcal{Y}} c(x,y)d\pi(y|x) + \epsilon\mathcal{R}(\pi(\cdot|x))$, where $\epsilon > 0$ and $\mathcal{R} : \mathcal{P}(\mathcal{Y}) \to \mathbb{R}$ is some functional (a.k.a. regularizer), e.g., variance [3, 35], kernel variance [34] or entropy [3]. With mild assumptions on the weak cost function $C$, an OT plan $\pi^*$ in (2) exists. We say that the family of its conditional distributions $\{\pi^*(\cdot|x)\}_{x\in\mathcal{X}}$ is the *conditional OT plan*.

**Entropic OT formulation** [13, 20]. It is common to consider entropy-based regularizers for (1):

$$\begin{cases} \text{EOT}^{(1)}_{c,\epsilon}(\mathbb{P}_0, \mathbb{P}_1) \\ \text{EOT}^{(2)}_{c,\epsilon}(\mathbb{P}_0, \mathbb{P}_1) \\ \text{EOT}_{c,\epsilon}(\mathbb{P}_0, \mathbb{P}_1) \end{cases} \stackrel{\text{def}}{=} \min_{\pi \in \Pi(\mathbb{P}_0,\mathbb{P}_1)} \int_{\mathcal{X}\times\mathcal{Y}} c(x,y)\pi(x,y) + \begin{cases} +\epsilon\text{KL}\left(\pi\|\mathbb{P}_0\times\mathbb{P}_1\right), & (3) \\ -\epsilon H(\pi), & (4) \\ -\epsilon\int_{\mathcal{X}} H\left(\pi(\cdot|x)\right)d\mathbb{P}_0(x). & (5) \end{cases}$$

Here KL is the Kullback–Leibler divergence and $H$ is the differential entropy, i.e., the minus KL divergence with the Lebesgue measure. Since $\pi \in \Pi(\mathbb{P}_0, \mathbb{P}_1)$, it holds that KL $(\pi\|\mathbb{P}_0\times\mathbb{P}_1) = H(\mathbb{P}_0) - \int_{\mathcal{X}} H(\pi(y|x))d\mathbb{P}_0(x) = -H(\pi) + H(\mathbb{P}_0) + H(\mathbb{P}_1)$, i.e., these formulations are equal up to an additive constant when $\mathbb{P}_0 \in \mathcal{P}_{ac}(\mathcal{X})$ and $\mathbb{P}_1 \in \mathcal{P}_{ac}(\mathcal{Y})$ and have finite entropy. Here we introduce "*ac*" subscript to indicate the subset of absolutely continuous distributions. With mild assumptions on $c, \mathbb{P}_0, \mathbb{P}_1$, the minimizer $\pi^*$ exists, it is **unique** and called the entropic OT plan. It is important to note that *entropic OT* (5) *is a case of weak OT* (2). Indeed, for the weak cost

$$C_{c,\epsilon}(x, \pi(\cdot|x)) \stackrel{\text{def}}{=} \int_{\mathcal{Y}} c(x,y)d\pi(y|x) - \epsilon H(\pi(\cdot|x)), \tag{6}$$

formulation (2) immediately turns to (5). This allows us to apply the theory of weak OT to EOT.

**Dual OT formulation.** There exists a wide range of dual formulations of OT [53, 48], WOT [2, 22] and EOT [20, 44]. We only recall the particular dual form for WOT from [3, 2] which serves as the main theoretical ingredient for our paper. For technical reasons, from now on we consider only $\mathbb{P}_1 \in \mathcal{P}_p(\mathcal{Y}) \subset \mathcal{P}(\mathcal{Y})$ for some $p \geq 1$, where subscript "$p$" indicates distributions with a finite $p$-th moment. We also assume that the weak cost $C : \mathcal{X} \times \mathcal{P}_p(\mathcal{Y}) \to \mathbb{R} \cup \{+\infty\}$ is lower bounded, *convex* in the second argument and jointly lower-semicontinuous in $\mathcal{X} \times \mathcal{P}_p(\mathcal{Y})$. In this case, a minimizer $\pi^*$ of WOT (2) exists [3, Theorem 3.2] and the following dual formulation holds [3, Eq. 3.3]:

$$\text{WOT}_C(\mathbb{P}_0, \mathbb{P}_1) = \sup_f \left\{ \int_{\mathcal{X}} f^C(x)d\mathbb{P}_0(x) + \int_{\mathcal{Y}} f(y)d\mathbb{P}_1(y) \right\}, \tag{7}$$

where $f \in \mathcal{C}_p(\mathcal{Y}) \stackrel{\text{def}}{=} \{f : \mathcal{Y} \to \mathbb{R}$ continuous s.t. $\exists \alpha, \beta \in \mathbb{R} : |f(\cdot)| \leq \alpha\|\cdot\|^p + \beta\}$ and $f^C$ is the so-called *weak C-transform* of $f$ which is defined by

$$f^C(x) \stackrel{\text{def}}{=} \inf_{\nu\in\mathcal{P}_p(\mathcal{Y})} \{C(x,\nu) - \int_{\mathcal{Y}} f(y)d\nu(y)\}. \tag{8}$$

Function $f$ in (7) is typically called the dual variable or the *Kantorovich potential*.

**SB problem with Wiener prior** [38, 11]. Let $\Omega$ be the space of $\mathbb{R}^D$-valued functions of time $t \in [0, 1]$ describing trajectories in $\mathbb{R}^D$, which start at time $t = 0$ and end at time $t = 1$. We use $\mathcal{P}(\Omega)$ to denote the set of probability distributions on $\Omega$, i.e., stochastic processes.

Consider two distributions $\mathbb{P}_0 \in \mathcal{P}_{2,ac}(\mathcal{X})$ and $\mathbb{P}_1 \in \mathcal{P}_{2,ac}(\mathcal{Y})$ with finite entropy. Let $\mathcal{F}(\mathbb{P}_0, \mathbb{P}_1) \subset \mathcal{P}(\Omega)$ be the subset of processes which have marginals $\mathbb{P}_0$ and $\mathbb{P}_1$ at times $t = 0$ and $t = 1$, respectively. Let $dW_t$ be the differential of the standard $\mathbb{R}^D$-valued Wiener process. Let $W^\epsilon \in \mathcal{P}(\Omega)$ be the Wiener process with the variance $\epsilon > 0$ which starts at $\mathbb{P}_0$ at time $t = 0$. It can be represented via the following stochastic differential equation (SDE): $dX_t = \sqrt{\epsilon}dW_t$ with $X_0 \sim \mathbb{P}_0$.

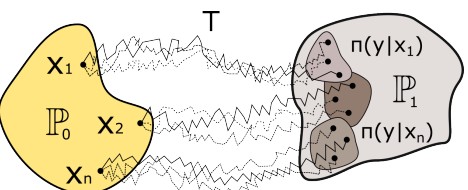

Figure 2: The bridge of Schrödinger.

The Schrödinger Bridge problem with the **Wiener prior** is the following:

$$\inf_{T\in\mathcal{F}(\mathbb{P}_0,\mathbb{P}_1)} \text{KL}\left(T\|W^\epsilon\right). \tag{9}$$

The inf is attained uniquely at some process $T^*$ [38, Proposition 4.1]. This process turns out to be a **diffusion** process and can be (uniquely) represented as the following SDE:

$$T^* : dX_t = v^*(X_t, t)dt + \sqrt{\epsilon}dW_t, \tag{10}$$

where $v^* : \mathbb{R}^D \times [0,1] \to \mathbb{R}^D$ is its drift function which we call the *optimal drift*. Hence, in (9), one may consider only diffusion processes $\subset \mathcal{F}(\mathbb{P}_0, \mathbb{P}_1)$ with the volatility $\epsilon$ coinciding with the volatility of the Wiener prior $W^\epsilon$. In turn, solving SB can be viewed as finding the optimal drift $v^*$.

**Link between SB and EOT problem.** The process $T^*$ solving SB (9) is related to the solution $\pi^*$ of EOT problem (5) *with the quadratic cost function* $c(x,y) = \frac{1}{2}\|x-y\|^2$. We start with some notations. For a process $T \in \mathcal{P}(\Omega)$, denote the joint distribution at time moments $t = 0, 1$ by $\pi^T \in \mathcal{P}(\mathcal{X} \times \mathcal{Y})$. Let $T_{|x,y}$ be the distribution of $T$ for $t \in (0,1)$ conditioned on $T$'s values $x, y$ at $t = 0, 1$.

> For the solution $T^*$ of SB (9), it holds that $\pi^{T^*} = \pi^*$, where $\pi^*$ is the EOT plan solving (5). Moreover, $T^*_{|x,y} = W^\epsilon_{|x,y}$, i.e., informally, the "*inner*" part of $T^*$ matches that of the prior $W^\epsilon$.

Conditional process $W^\epsilon_{|x,y}$ is well-known as the **Brownian Bridge**. Due to this, given $x, y$, simulating the trajectories of $W^\epsilon_{|x,y}$ is rather straightforward. Thanks to this aspect, SB and EOT can be treated as *nearly* equivalent problems. Still EOT solution $\pi^*$ does not directly yield the optimal drift $v^*$. However, it is known that the density $\frac{d\pi^*(x,y)}{d(x,y)}$ of $\pi^*$ has the specific form [38, Theorem 2.8], namely, $\frac{d\pi^*(x,y)}{d(x,y)} = \widetilde{\varphi}^*(x)\mathcal{N}(y|x, \epsilon I)\varphi^*(y)$, where functions $\varphi^*, \widetilde{\varphi}^* : \mathbb{R}^D \to \mathbb{R}$ are called the *Schrödinger potentials*. From this equality one gets the expression for $\varphi^*(\cdot)$ and the density of $\pi^*(\cdot|x)$:

$$\frac{d\pi^*(y|x)}{dy} \propto \mathcal{N}(y|x, \epsilon I)\varphi^*(y) \qquad \Longrightarrow \qquad \varphi^*(y) \propto \frac{d\pi^*(y|x)}{dy} \cdot \left[\mathcal{N}(y|x, \epsilon I)\right]^{-1} \quad (11)$$

up to multiplicative constants. One may recover the optimal drift $v^*$ via [38, Proposition 4.1]

$$v^*(x,t) = \epsilon \nabla \log \int_{\mathbb{R}^D} \mathcal{N}(y|x, (1-t)\epsilon I_D)\varphi^*(y)dy. \qquad (12)$$

Here the normalization constant vanishes when one computes $\nabla \log(\cdot)$. Thus, technically, knowing the (unnormalized) density of $\pi^*$, one may recover the optimal drift $v^*$ for SB (9).

## 2  Background: Solving Continuous OT and SB Problems

Although OT (2), EOT (5) and SB (9) problems are well-studied in theory, solving them in practice is challenging. Existing OT solvers are of two main types: *discrete* [44] and *continuous* [32]. **Our benchmark is designed for continuous EOT solvers**; discrete OT/EOT is out of the scope of the paper.

Continuous OT assumes that distributions $\mathbb{P}_0$ and $\mathbb{P}_1$ are continuous and accessible only via their random samples $X = \{x_1, \ldots, x_N\} \sim \mathbb{P}_0$ and $Y = \{y_1, \ldots, y_M\} \sim \mathbb{P}_1$. The goal is to recover an OT plan $\pi^*$ between *entire* $\mathbb{P}_0$ and $\mathbb{P}_1$ but using only $X$ and $Y$. Most continuous OT solvers do this via employing neural networks to implicitly learn the conditional distributions $\widehat{\pi}(\cdot|x) \approx \pi^*(\cdot|x)$. In turn, SB solvers learn the optimal drift $\widehat{v} \approx v^*$ but it is anyway used to produce samples $y \sim \widehat{\pi}(\cdot|x)$ via solving SDE $dX_t = \widehat{v}(x,t)dt + \sqrt{\epsilon}dW_t$ starting from $X_0 = x$ (sampled from $\mathbb{P}_0$) at time $t = 0$.

After training on available samples $X$ and $Y$, continuous solvers may produce $y \sim \widehat{\pi}(\cdot|x_{\text{test}})$ for previously unseen samples $x_{\text{test}} \sim \mathbb{P}_0$. This is usually called the out-of-sample estimation. It allows applying continuous OT solver to generative modelling problems such as the **image synthesis** (*noise-to-data*) and **translation** (*data-to-data*). In both these cases, $\mathbb{P}_1$ is a data distribution, and $\mathbb{P}_0$ is either a noise (in synthesis) or some other data distribution (in translation). Many recent OT solvers achieve competitive performance in synthesis [12, 15, 46] and translation [35, 34] tasks.

Continuous OT/SB solvers are usually referred to as **neural OT/SB** because they employ neural networks. There exist a lot of neural OT solvers for classic OT (1) [46, 17, 56, 40, 30, 19], see also [32, 31] for **surveys**, weak OT (2) [34, 35, 1], entropic OT (5) [49, 14, 42] and SB (9) [52, 15, 9, 23]. Providing a concise but still explanatory overview of them is nearly impossible as the *underlying principles of many of them are rather different and non-trivial*. We list only EOT/SB solvers which are relevant to our benchmark and provide a brief summary of them in Table 1. In §5, we test all these solvers on our continuous benchmark distributions which we construct in subsequent §3.

**Approaches to evaluate solvers.** As seen from Table 1, each paper usually tests its solver on a restricted set of examples which rarely intersects with those from the other papers. In particular, some papers consider *data→data* tasks, while the others focus on *noise→data*. Due to this, there is

| | Solver | Underlying principle and parameterization | Evaluated as EOT/SB | Tested in generation (*noise→data*) | Tested in translation (*data→data*) |
|---|---|---|---|---|---|
| **EOT** solvers | LSOT [49, 21] | Solves classic dual EOT [20, §3.1] with 2 NNs. Learns 1 more NN for the barycentric projection. | ✗ | MNIST (32x32) | MNIST→USPS(16x16), USPS→MNIST (16x16), SVHN→MNIST (3x32x32) |
| | SCONES [14] | Combines LSOT's potentials with a score model for $\mathbb{P}_1$ to sample from $\pi^*(\cdot|x)$ via Langevin dynamics. | Gaussians | ✗ | CelebA Upscale (**3x64x64**) |
| | NOT* [35] | Solves max-min reformulation of weak OT dual (7) with 2 NNs (transport map and potential). | colspan: * This is a generic neural solver for weak OT but it has **not** been tested with the **entropic** cost function. | | |
| | EGNOT [42] | Employs energy-based modeling (EBM [37]) to solve weak EOT dual (7); non-minimax; 1NN. | Gaussians | ✗ | Colored MNIST 2→3 (3x32x32) |
| **SB** solvers | ENOT [23] | Solves max-min reformulation of SB with 2 NNs (potential and SDE drift). | Gaussians | ✗ | CelebA Upscale (**3x64x64**), Colored MNIST 2→3 (3x32x32) |
| | MLE-SB [52] | Alternate solving of two Half Bridge (HB) problems. HB is solved via drift estimation with GP [55]. | ✗ | ✗ | Single Cell data ($D = 5$), Motion Capture ($D = 4$) |
| | DiffSB [15] | Iterative Mean-Matching Proportional Fitting 2 NNs for forward and backward SDE drifts | Gaussians | CelebA (32x32) | MNIST→EMNIST (32x32) |
| | FB-SDE [9] | Likelihood training of SB 2 NNs for the $\nabla$ log of Schrödinger potentials | ✗ | MNIST (32x32), CelebA (3x32x32), CIFAR-10 (3x32x32) | ✗ |

Table 1: Table of existing continuous (neural) solvers for EOT/SB.

no clear understanding of the superiority of one solver over the other. Importantly, in many cases, the *quatitative* evaluation is done exclusively via the metrics of the downstream task. For example, [15, 9, 23, 14] consider image generation or translation tasks and test the quality of generated images via FID [24]. That is, they compare generated *marginal* distribution $\widehat{\pi}_1$ with target $\mathbb{P}_1$. This allows to access the generative performance of solvers but gives **no hint whether they actually learn the true EOT/SB solution**. Works [14, 23, 42, 15] do a step toward opening the veil of secrecy and test their solvers in the Gaussian case. Unfortunately, it is rather trivial and may not be representative.

## 3 Constructing Benchmark Pairs for OT and SB: Theory

In this section, we present our theoretical results allowing us to construct pairs of distributions with the EOT/SB solution known by the construction. We provide proofs in Appendix A.

### 3.1 Generic Optimal Transport Benchmark Idea

For a given distribution $\mathbb{P}_0 \in \mathcal{P}(\mathcal{X})$, we want to construct a distribution $\mathbb{P}_1 \in \mathcal{P}_p(\mathcal{Y})$ such that some OT plan $\pi^* \in \Pi(\mathbb{P}_0, \mathbb{P}_1)$ for a given weak OT cost function $C$ between them is known by the construction. That is, $\pi_0^* = \mathbb{P}_0$, $\pi_1^* = \mathbb{P}_1$ and $\pi^*$ minimizes (2). In this case, $(\mathbb{P}_0, \mathbb{P}_1)$ may be used as a **benchmark pair** with a known OT solution. Our following main theorem provides a way to do so.

**Theorem 3.1** (Optimal transport benchmark constructor). *Let $\mathbb{P}_0 \in \mathcal{P}(\mathcal{X})$ be a given distribution, $f^* \in \mathcal{C}_p(\mathcal{Y})$ be a given function and $C : \mathcal{X} \times \mathcal{P}_p(\mathcal{Y}) \to \mathbb{R}$ be a given jointly lower semi-continuous, convex in the second argument and lower bounded weak cost. Let $\pi^* \in \mathcal{P}(\mathcal{X} \times \mathcal{Y})$ be a distribution for which $\pi_0^* = \mathbb{P}_0$ and for all $x \in \mathcal{X}$ it holds that*

$$\pi^*(\cdot|x) \in \operatorname*{arginf}_{\mu \in \mathcal{P}_p(\mathcal{Y})} \{C(x, \mu) - \int_{\mathcal{Y}} f^*(y) d\mu(y)\}. \tag{13}$$

*Let $\mathbb{P}_1 \stackrel{def}{=} \pi_1^*$ be the second marginal of $\pi^*$ and assume that $\mathbb{P}_1 \in \mathcal{P}_p(\mathcal{Y})$. Then $\pi^*$ is an **OT plan** between $\mathbb{P}_0$ and $\mathbb{P}_1$ (it minimizes (2)) and $f^*$ is an **optimal dual potential** (it maximizes (7)).*

Thanks to our theorem, given a pair $(\mathbb{P}_0, f^*) \in \mathcal{P}(\mathcal{X}) \times \mathcal{C}_p(\mathcal{Y})$ of a distribution and a potential, one may produce a distribution $\mathbb{P}_1$ for which an OT plan between them is known by the construction. This may be done by picking $\pi^* \in \Pi(\mathbb{P})$ whose conditionals $\pi^*(\cdot|x)$ minimize (13).

While our theorem works for rather general costs $C$, it may be non-trivial to compute a minimizer $\pi^*(\cdot|x)$ in the weak $C$-transform (8), e.g., to sample from it or to estimate its density. Also, we note that our theorem states that $\pi^*$ is optimal but does not claim that it is the unique OT plan. These aspects may complicate the usage of the theorem for constructing the benchmark pairs $(\mathbb{P}_0, \mathbb{P}_1)$ for general costs $C$. Fortunately, both these issues vanish when we consider EOT, see below.

### 3.2 Entropic Optimal Transport Benchmark Idea

For $C = C_{c,\epsilon}$ (6) with $\epsilon > 0$, the characterization of minimizers $\pi^*(\cdot|x)$ in (13) is almost explicit.

**Theorem 3.2** (Entropic optimal transport benchmark constructor). *Let $\mathbb{P}_0 \in \mathcal{P}(\mathcal{X})$ be a given distribution and $f^* \in \mathcal{C}_p(\mathcal{Y})$ be a given potential. Assume that $c : \mathcal{X} \times \mathcal{Y} \to \mathbb{R}$ is lower bounded and $(x, \mu) \mapsto \int_{\mathcal{Y}} c(x,y)d\mu(y)$ is lower semi-continuous in $\mathcal{X} \times \mathcal{P}_p(\mathcal{Y})$. Furthermore, assume that there exists $M \in \mathbb{R}_+$ such that for all $x \in \mathcal{X}$ it holds that $M_x \stackrel{\text{def}}{=} \int_{\mathcal{Y}} \exp\left(-\frac{c(x,y)}{\epsilon}\right) dy \leq M$. Assume that for all $x \in \mathcal{X}$ value $Z_x \stackrel{\text{def}}{=} \int_{\mathcal{Y}} \exp\left(\frac{f^*(y)-c(x,y)}{\epsilon}\right) dy$ is finite. Consider the joint distribution $\pi^* \in \mathcal{P}(\mathcal{X} \times \mathcal{Y})$ whose first marginal distribution satisfies $\pi_0^* = \mathbb{P}_0$ and for all $x \in \mathcal{X}$ it holds that*

$$\frac{d\pi^*(y|x)}{dy} = \frac{1}{Z_x} \exp\left(\frac{f^*(y)-c(x,y)}{\epsilon}\right) \tag{14}$$

*and $\pi^*(\cdot|x) \in \mathcal{P}_p(\mathcal{Y})$. Then if $\mathbb{P}_1 \stackrel{\text{def}}{=} \pi_1^*$ belongs to $\mathcal{P}_p(\mathcal{Y})$, the distribution $\pi^*$ is an **EOT plan** for $\mathbb{P}_0, \mathbb{P}_1$ and cost $C_{c,\epsilon}$. Moreover, if $\int_{\mathcal{X}} C_{c,\epsilon}(x, \pi^*(\cdot|x)) d\mathbb{P}_0(x) < \infty$, then $\pi^*$ is the **unique** EOT plan.*

Our result above requires some technical assumptions on $c$ and $f^*$ but a reader should not worry as they are easy to satisfy in popular cases such as the quadratic cost $c(x, y) = \frac{1}{2}\|x - y\|^2$ (§3.3). The important thing is that our result allows **sampling** $y \sim \pi^*(\cdot|x)$ from the conditional EOT plan by using MCMC methods [5, §11.2] since (14) provides the **unnormalized density** of $\pi^*(y|x)$. Such sampling may be time-consuming, which is why we provide a clever approach to avoid MCMC below.

### 3.3 Fast Sampling With LogSumExp Quadratic Potentials.

In what follows, we propose a way to overcome the challenging sampling problem by considering the case $c(x, y) = \frac{\|x - y\|^2}{2}$ and the special family of functions $f^*$. For brevity, for a matrix $A \in \mathbb{R}^{D \times D}$ and $b \in \mathbb{R}^D$, we introduce $\mathcal{Q}(y|b, A) \stackrel{\text{def}}{=} \exp\left[-\frac{1}{2}(y - b)^T A(y - b)\right]$. Henceforth, we choose the potential $f^*$ to be a weighted log-sum-exp (LSE) of $N$ quadratic functions:

$$f^*(y) \stackrel{\text{def}}{=} \epsilon \log \sum_{n=1}^{N} w_n \mathcal{Q}(y|b_n, \epsilon^{-1} A_n) \tag{15}$$

Here $w_n \geq 0$ and we put $A_n$ to be a symmetric matrix with eigenvalues in range $(-1, +\infty)$. We say that such potentials $f^*$ are **appropriate**. One may also check that $f^* \in \mathcal{C}_2(\mathcal{Y})$ as it is just the LSE smoothing of quadratic functions. Importantly, for this potential $f$ and the quadratic cost, $\pi^*(\cdot|x)$ is a Gaussian mixture, from which one can efficiently sample **without using MCMC methods**.

**Proposition 3.3** (Entropic OT solution for LSE potentials). *Let $f^*$ be a given appropriate LSE potential (15) and let $\mathbb{P}_0 \in \mathcal{P}_2(\mathcal{X}) \subset \mathcal{P}(\mathcal{X})$. Consider the plan $d\pi^*(x, y) = d\pi^*(y|x) d\mathbb{P}_0(x)$, where*

$$\frac{d\pi^*(y|x)}{dy} = \sum_{n=1}^{N} \gamma_n \mathcal{N}(y|\mu_n(x), \Sigma_n) \text{ with } \Sigma_n \stackrel{\text{def}}{=} \epsilon(A_n + I)^{-1}, \mu_n(x) \stackrel{\text{def}}{=} (A_n + I)^{-1}(A_n b_n + x),$$

$$\gamma_n \stackrel{\text{def}}{=} \widetilde{w}_n / \sum_{n=1}^{N} \widetilde{w}_n, \qquad \widetilde{w}_n \stackrel{\text{def}}{=} w_n (2\pi)^{\frac{D}{2}} \sqrt{\det(\Sigma_n)} \mathcal{Q}(x|b_n, \frac{1}{\epsilon} I - \frac{1}{\epsilon^2} \Sigma_n).$$

*Then it holds that $\mathbb{P}_1 \stackrel{\text{def}}{=} \pi_1^*$ belongs to $\mathcal{P}_2(\mathcal{Y})$ and the joint distribution $\pi^*$ is the **unique EOT plan** between $\mathbb{P}_0$ and $\mathbb{P}_1$ for cost $C_{c,\epsilon}$ with $c(x, y) = \frac{1}{2}\|x - y\|^2$.*

We emphasize that although each conditional distribution $\pi^*(\cdot|x)$ is a Gaussian mixture, in general, this **does not** mean that $\pi^*$ or $\mathbb{P}_1 = \pi_1^*$ is a Gaussian mixture, even when $\mathbb{P}_0$ is Gaussian. This aspect does not matter for our construction, and we mention it only for the completeness of the exposition.

### 3.4 Schrödinger Bridge Benchmark Idea

Since there is a link between EOT and SB, our approach allows us to immediately obtain a solution to the Schrödinger Bridge between $\mathbb{P}_0$ and $\mathbb{P}_1$ (constructed with an LSE potential $f^*$).

**Corollary 3.4** (Solution for SB between $\mathbb{P}_0$ and constructed $\mathbb{P}_1$). *In the context of Theorem 3.2, let $p = 2$ and consider $c(x, y) = \frac{1}{2}\|x - y\|^2$. Assume that $\mathbb{P}_0 \in \mathcal{P}_{2,ac}(\mathcal{X}) \subset \mathcal{P}(\mathcal{X})$ and both $\mathbb{P}_0$ and $\mathbb{P}_1$ (constructed with a given $f^*$) have finite entropy. Then it holds that $\varphi^*(y) \stackrel{\text{def}}{=} \exp\left(\frac{f^*(y)}{\epsilon}\right)$ is a Schrödinger potential providing the **optimal drift** $v^*$ for SB via formula (12).*

Although the drift is given in the closed form, its computation may be challenging, especially in high dimensions. Fortunately, as well as for EOT, for the quadratic cost $c(x, y) = \frac{\|x-y\|^2}{2}$ and our LSE (15) potentials $f^*$, we can derive the optimal drift explicitly.

| $\epsilon$ | Metric | EOT solvers | | | | SB solvers | | | | |
|---|---|---|---|---|---|---|---|---|---|---|
| | | ⌊LSOT⌉ | ⌊SCONES⌉ | ⌊NOT⌉ | ⌊EgNOT⌉ | ⌊ENOT⌉ | ⌊MLE-SB⌉ | ⌊DiffSB⌉ | ⌊FB-SDE-A⌉ | ⌊FB-SDE-J⌉ |
| 0.1 | $\mathbb{BW}_2^2$-UVP | ✗ Do not work for small $\epsilon$ due to numerical instability [14, §5.1]. | | 😊 | 😊 | 😐 | 😐 | ☹️ | ☹️ | 😊 |
| | $\mathrm{c}\mathbb{BW}_2^2$-UVP | | | 😊 | 😐 | 😐 | 😊 | 😐 | ☹️ | ☹️ |
| 1 | $\mathbb{BW}_2^2$-UVP | ☹️ | ☹️ | 😐 | 😊 | 😐 | 😐 | 😐 | 😐 | 😊 |
| | $\mathrm{c}\mathbb{BW}_2^2$-UVP | ☹️ | ☹️ | 😐 | ☹️ | 😐 | 😊 | ☹️ | ☹️ | ☹️ |
| 10 | $\mathbb{BW}_2^2$-UVP | ☹️ | ☹️ | 😐 | 😊 | ☹️ | 😐 | ✗ Diverge with the default hyperparameters. May require more hyperparameter tuning. | | |
| | $\mathrm{c}\mathbb{BW}_2^2$-UVP | ☹️ | ☹️ | ☹️ | ☹️ | ☹️ | ☹️ | | | |

Table 2: The summary of EOT/SB solvers' quantitative performance in $\mathrm{c}\mathbb{BW}_2^2$-UVP and $\mathbb{BW}_2^2$-UVP metrics on our mixtures pairs. Detailed evaluation and coloring principles are given in Appendix B.

**Corollary 3.5** (SB solution for LSE potentials). *Let $f^*$ be a given appropriate LSE potential* (15) *and consider a distribution $\mathbb{P}_0 \in \mathcal{P}_{2,ac}(\mathcal{X})$ with finite entropy. Let $\mathbb{P}_1$ be the one constructed in Proposition 3.3. Then it holds that $\mathbb{P}_1$ has finite entropy, belongs to $\mathcal{P}_{2,ac}(\mathcal{Y})$ and*

$$v^*(x,t) = \nabla_x \log \sum_{n=1}^{N} w_n \sqrt{\det(\Sigma_n^t)} \mathcal{Q}(x|b_n, \frac{1}{\epsilon(1-t)}I - \frac{1}{\epsilon^2(1-t)}\Sigma_n^t) \qquad (16)$$

*is the optimal drift for the SB between $\mathbb{P}_0$ and $\mathbb{P}_1$. Here $A_n^t \stackrel{def}{=} (1-t)A_n$ and $\Sigma_n^t \stackrel{def}{=} \epsilon(A_n^t + I)^{-1}$.*

# 4  Constructing Benchmark Pairs for OT and SB: Implementation

Our benchmark is implemented using `PyTorch` framework and is publicly available at

https://github.com/ngushchin/EntropicOTBenchmark

It provides code to sample from our constructed continuous benchmark pairs $(\mathbb{P}_0, \mathbb{P}_1)$ for various $\epsilon$, see §4.1 for details of these **pairs**. In §4.2, we explain the **intended usage** of these pairs.

## 4.1  Constructed Benchmark Pairs
We construct various pairs in dimensions $D$ up to 12288 and $\epsilon \in \{0.1, 1, 10\}$. Our mixtures pairs simulate *noise→data* setup and images pairs simulate *data→data* case.

**Mixtures benchmark pairs.** We consider EOT with $\epsilon \in \{0.1, 1, 10\}$ in space $\mathbb{R}^D$ with dimension $D \in \{2, 16, 64, 128\}$. We use a centered Gaussian as $\mathbb{P}_0$ and we use LSE function (15) with $N = 5$ for constructing $\mathbb{P}_1$ (Proposition 3.3). In this case, the constructed distribution $\mathbb{P}_1$ has 5 modes (Fig. 3a, 3f). Details of particular parameters $(A_n, b_n, w_n,$ etc.) are given in Appendix B.

**Images benchmark pairs.** We consider EOT with $\epsilon \in \{0.1, 1, 10\}$. As distribution $\mathbb{P}_0$, we use the approximation of the distribution of $64 \times 64$ RGB images ($D = 12288$) of CelebA faces dataset [39]. Namely, we train a normalizing flow with Glow architecture [29]. It is absolutely continuous by the construction and allows straightforward sampling from $\mathbb{P}_0$. For constructing distribution $\mathbb{P}_1$, we also use LSE function $f^*$. We fix $N = 100$ random samples from $\mathbb{P}_0$ for $b_n$ and choose all $A_n \equiv I$. Details of $w_n$ are given in Appendix C. For these parameters, samples from $\mathbb{P}_1$ look like noised samples from $\mathbb{P}_0$ which are shifted to one of $\mu_n$ (Fig. 4).

By the construction of distribution $\mathbb{P}_1$, obtaining the conditional EOT plan $\pi^*(y|x)$ between $(\mathbb{P}_0, \mathbb{P}_1)$ may be viewed as learning the *noising* model. From the practical perspective, this looks less interesting than learning the *de-noising* model $\pi^*(x|y)$. Due to this, working with images in §5, we always test EOT solvers in $\mathbb{P}_1 \to \mathbb{P}_0$ direction, i.e., recovering $\pi^*(x|y)$ and generating clean samples $x$ form noised $y$. The reverse conditional OT plans $\pi^*(x|y)$ are not as tractable as $\pi^*(y|x)$. We overcome this issue with MCMC in the latent space of the normalizing flow (Appendix C).

## 4.2  Intended Usage of the Benchmark Pairs
The EOT/SB solvers (§2) provide an approximation of the conditional OT plan $\widehat{\pi}(\cdot|x) \approx \pi^*(\cdot|x)$ from which one can sample (given $x \sim \mathbb{P}_0$). In particular, SB solvers recover the approximation of the optimal drift $\widehat{v} \approx v^*$; it is anyway used to produce samples $y \sim \widehat{\pi}(\cdot|x)$ via solving SDE $dX_t = \widehat{v}(x,t)dt + \sqrt{\epsilon}dW_t$ starting from $X_0 = x$ at time $t = 0$. Therefore, **the main goal of our benchmark** is to provide a way to compare such approximations $\widehat{\pi}, \widehat{v}$ with the ground truth. *Prior to our work, this was not possible* due to the lack of non-trivial pairs $(\mathbb{P}_0, \mathbb{P}_1)$ with known $\pi^*, v^*$.

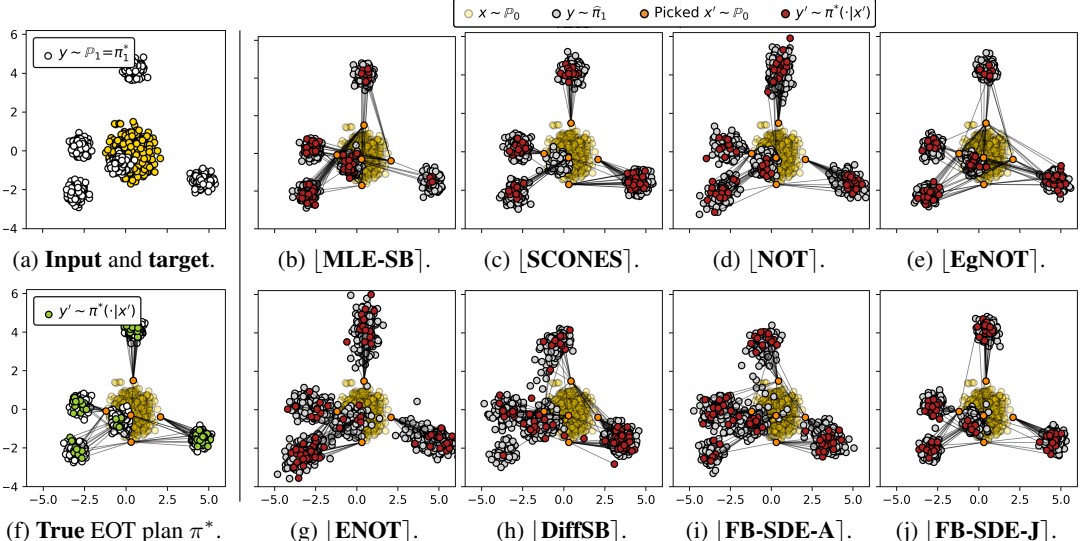

Figure 3: Qualitative results of EOT/SB solvers on our mixtures benchmark pair with $(D, \epsilon) = (16, 1)$. The distributions are visualized using 2 PCA components of target distribution $\mathbb{P}_1$. Additional examples of performance on pairs with other $\epsilon \in \{0.1, 10\}$ are given in Appendix B.

For each of the constructed pairs $(\mathbb{P}_0, \mathbb{P}_1)$, we provide the code to do 5 main things: **(a)** sample $x \sim \mathbb{P}_0$; **(b)** sample $y \sim \pi^*(\cdot|x)$ for any given $x$; **(c)** sample pairs $(x, y) \sim \pi^*$ from the EOT plan; **(d)** sample $y \sim \mathbb{P}_1$; **(e)** compute the optimal drift $v^*(x, t)$. Function **(c)** is just a combination of **(a)** and **(b)**. For images pairs we implement the extra functionality **(f)** to sample $x \sim \pi^*(\cdot|y)$ using MCMC. Sampling **(d)** is implemented via discarding $x$ (not returning it to a user) in $(x, y)$ in **(c)**.

When **training** a neural EOT/SB solver, one should use random batches from **(a,d)**. Everything coming from (**b,c,e,f**) should be considered as **test** information and used only for evaluation purposes. Also, for each of the benchmark pairs (mixtures and images), we provide a hold-out **test** for evaluation.

## 5   Experiments: Testing EOT and SB Solvers on Our Benchmark Pairs

Now we train various existing EOT/SB solvers from Table 1 on our benchmark pairs $(\mathbb{P}_0, \mathbb{P}_1)$ to showcase how well they capture the ground truth EOT plan. For solvers' details, see Appendix D.

MIXTURES BENCHMARK PAIRS. For quantitative analysis, we propose the following metric:

$$\mathrm{cBW}_2^2\text{-UVP}\big(\widehat{\pi}, \pi^*\big) \stackrel{\text{def}}{=} \frac{100\%}{\frac{1}{2}\mathrm{Var}(\mathbb{P}_1)} \int_{\mathcal{X}} \mathrm{BW}_2^2\big(\widehat{\pi}(\cdot|x), \pi^*(\cdot|x)\big) d\mathbb{P}_0(x). \tag{17}$$

For each $x$ we compare **conditional** distributions $\widehat{\pi}(\cdot|x)$ and $\pi^*(\cdot|x)$ with each other by using the Bures-Wasserstein metric [16], i.e., the Wasserstein-2 distance between Gaussian approximations of distributions. Then we average this metric w.r.t. $x \sim \mathbb{P}_0$. The final normalization $\frac{1}{2}\mathrm{Var}(\mathbb{P}_1)$ is chosen so that the trivial baseline which maps the entire $\mathbb{P}_0$ to the mean of $\mathbb{P}_1$ provides $100\%$ error. Metric (17) is a modification of the standard $\mathrm{BW}_2^2$-UVP [14, 23, 42, 33, 18] for the conditional setting. For completeness, we also report the **standard** $\mathrm{BW}_2^2$-UVP to check how well $\widehat{\pi}_1$ matches $\pi_1^*$.

**DISCLAIMER.** We found that most solvers' performance **significantly** depends on the selected hyper-parameters. We neither have deep knowledge of many solvers nor have the resources to tune them to achieve the best performance on our benchmark pairs. Thus, *we kindly invite the interested authors of solvers to improve the results for their solvers.* Meanwhile, we report the results of solvers with their default configs and/or with limited tuning. Nevertheless, we present a hyperparameter study in Appendix E to show that the chosen hyperparameters are a reasonable fit for the considered tasks. Our goal here is to find out and explain the issues of the methods which are due to their principle rather than non-optimal hyperparameter selection.

The detailed results are in Appendix B. Here we give their concise summary (Table 2) and give a qualitative example (Fig. 3) of solvers' performance on our mixtures pair with $(D, \epsilon) = (16, 1)$.

**EOT SOLVERS.** $\lceil$LSOT$\rceil$ and $\lceil$SCONES$\rceil$ solvers work only for medium/large $\epsilon = 1, 10$. $\lceil$LSOT$\rceil$ learns only the barycentric projection [49, Def. 1] hence naturally experiences large errors and even collapses (Fig. 3b). $\lceil$SCONES$\rceil$ solver works better but recovers the plan with a large error. We think

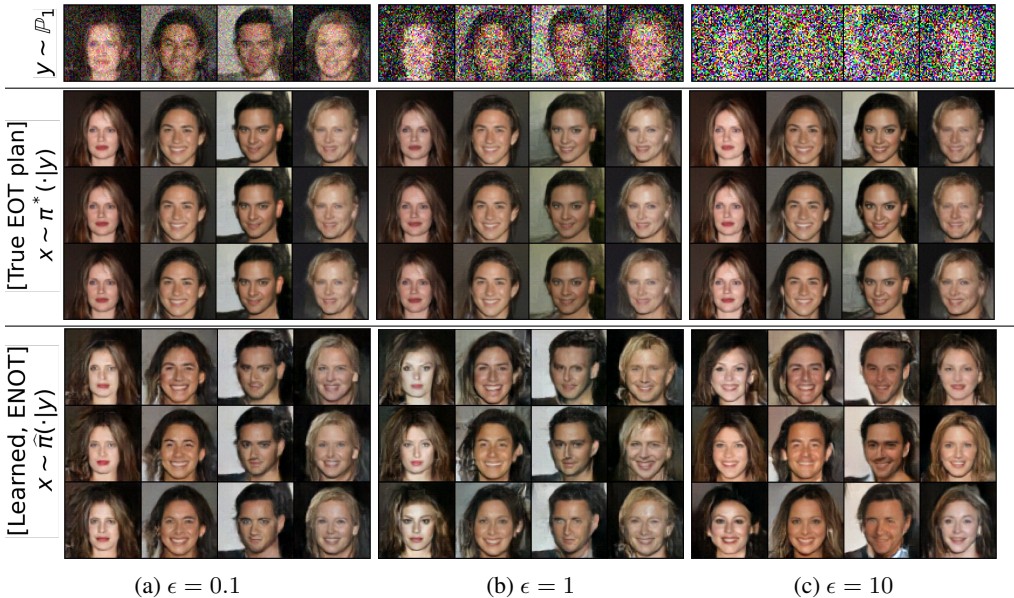

(a) $\epsilon = 0.1$                (b) $\epsilon = 1$                (c) $\epsilon = 10$

Figure 4: Qualitative comparison of ground truth samples $x \sim \pi^*(\cdot|y)$ with samples produced by $\lceil$**ENOT**$\rceil$. With the increase of $\epsilon$, the diversity increases but the precision of image restoration drops.

this is due to using the Langevin dynamic [14, Alg. 2] during the inference which gets stuck in modes plus the imprecise target density approximation which we employed (Appendix D). $\lceil$**EgNOT**$\rceil$ solver also employs Langevin dynamic and possibly experiences the same issue (Fig. 3e). Interestingly, our evaluation shows that it provides a better metric in matching the target distribution $\mathbb{P}_1$. $\lceil$**NOT**$\rceil$ was originally not designed for EOT because it is non-trivial to estimate entropy from samples. To fix this issue, we modify the authors' code for EOT by employing *conditional normalizing flow* (CNF) as the generator. This allows us to estimate the entropy from samples and hence apply the solver to the EOT case. It scores good results despite the restrictiveness of the used architecture (Fig. 3d).

**SB SOLVERS.** For $\lceil$**MLE-SB**$\rceil$, the original authors' implementation uses Gaussian processes as parametric approximators instead of neural nets [52]. Since other SB solvers ($\lceil$**DiffSB**$\rceil$, $\lceil$**FB-SDE-A(J)**$\rceil$ and $\lceil$**ENOT**$\rceil$) use neural nets, after discussion with the authors of $\lceil$**MLE-SB**$\rceil$, we decided to use neural nets in their solver as well. All SB solvers work reasonably well, but their performance drops as the $\epsilon$ increases. This is because it becomes more difficult to model the entire diffusion (with volatility $\epsilon$); these solvers may require more discretization steps. Still, the case of large $\epsilon$ is not very interesting since, in this case, the EOT is almost equal to the trivial independent plan $\mathbb{P}_0 \times \mathbb{P}_1$.

**IMAGES BENCHMARK PAIRS**. There are only two solvers which have been tested by the authors in their papers in such a large-scale *data→data* setup ($64 \times 64$ RGB images), see Table 1. Namely, these are $\lceil$**SCONES**$\rceil$ and $\lceil$**ENOT**$\rceil$. Unfortunately, we found that $\lceil$**SCONES**$\rceil$ yields unstable training on our benchmark pairs, probably due to too small $\epsilon$ for it, see [14, §5.1]. Therefore, we only report the results of $\lceil$**ENOT**$\rceil$ solver. For completeness, we tried to run $\lceil$**DiffSB**$\rceil$, $\lceil$**FB-SDE-A**$\rceil$ solvers with their configs from *noise→data* generative modelling setups but they diverged. We also tried $\lceil$**NOT**$\rceil$ with convolutional CNF as the generator but it also did not converge. We leave adapting these solvers for high-dimensional *data→data* setups for future studies. Hence, here we test only $\lceil$**ENOT**$\rceil$.

In Fig. 4, we qualitatively see that $\lceil$**ENOT**$\rceil$ solver *only for small $\epsilon$* properly learns the EOT plan $\pi^*$ and sufficiently well restores images from the input noised inputs. As there is anyway the lack of baselines in the field of neural EOT/SB, we plan to release these $\lceil$**ENOT**$\rceil$ checkpoints and expect them to become a **baseline for future works** in the field. Meanwhile, in Appendix C, we discuss possible metrics which we recommend to use to compare with these baselines.

## 6   Discussion

**Potential Impact.** Despite the considerable growth of the field of EOT/SB, there is still no standard way to test existing neural (continuous) solvers. In our work, we fill this gap. Namely, *we make a step towards bringing clarity and healthy competition to this research area by proposing the first-ever theoretically-grounded EOT/SB benchmark*. We hope that our constructed benchmark pairs will

become the standard playground for testing continuous EOT/SB solvers as part of the ongoing effort to advance computational OT/SB, in particular, in its application to generative modelling.

**Limitations (benchmark).** We employ LSE quadratic functions (15) as optimal Kantorovich potentials to construct benchmark pairs. It is unclear whether our benchmark sufficiently reflects the practical scenarios in which the EOT/SB solvers are used. Nevertheless, our methodology is generic and can be used to construct new benchmark pairs but may require MCMC to sample from them.

To show that the family of EOT plans which can be produced with LSE potentials is rich enough, we provide a heuristic recipe on how to construct benchmark pairs simulating given real-world datasets, see Appendix H. As we show there, the recipe works on several non-trivial single-cell datasets [36, 8]. Thus, we conjecture that LSE potentials may be sufficient to represent any complex distribution just like the well-celebrated Gaussian mixtures are capable of approximating any density [43]. We leave this inspiring theoretical question open for future studies.

For completeness, we note that our images benchmark pairs use LSE potentials and do not require MCMC for sampling from marginals $\mathbb{P}_0, \mathbb{P}_1$, i.e., to get clean and noisy images, respectively. However, for computing the test conditional FID (Appendix C) of EOT/SB solvers, MCMC is needed to sample clean images $x \sim \pi^*(\cdot|y)$ conditioned on noisy inputs $y$. This may introduce extra sources of error.

**Limitations (evaluation).** We employ $\mathbb{BW}_2^2$-UVP for the quantitative evaluation (§5) as it is popular in OT field [33, 14, 18, 23, 42] . However, it may not capture the full picture as it only compares the 1st and 2nd moments of distributions. We point to developing of novel evaluation metrics for neural OT/SB solvers as an important and helpful future research direction.

Following our disclaimer in §5, we acknowledge one more time, that the hyper-parameters tuning of the solvers which we test on our proposed benchmark is not absolutely comprehensive. It is possible that we might have missed something and did not manage to achieve the best possible performance in each particular case. At the same time, our Appendix E shows the extensive empirical study of the key hyper-parameters and it seems that the metrics reported are reasonably close to the optimal ones.

## 7 Acknowledgements

This work was partially supported by the Skoltech NGP Program (Skoltech-MIT joint project).

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

# A Proofs

*Proof of Theorem 3.1.* By the definition of $\mathbb{P}_1$, it holds that $\pi^* \in \Pi(\mathbb{P}_0, \mathbb{P}_1)$. It suffices to show that $\pi^*$ attains the optimal cost. Let $\mathbf{Cost}(\pi)$ be the value of weak OT functional for a plan $\pi$, i.e.,

$$\mathbf{Cost}(\pi) \stackrel{\text{def}}{=} \int_{\mathcal{X}} C(x, \pi(\cdot|x)) d\mathbb{P}(x).$$

We consider weak OT (2) between $\mathbb{P}_0 \in \mathcal{P}(\mathcal{X})$ and $\mathbb{P}_1 \in \mathcal{P}_p(\mathcal{X})$ and use its dual form (7):

$$\mathbf{Cost}(\mathbb{P}_0, \mathbb{P}_1) = \sup_f \left\{ \int_{\mathcal{X}} f^C(x) d\mathbb{P}_0(x) + \int_{\mathcal{Y}} f(y) d\mathbb{P}_1(y) \right\} =$$

$$\sup_f \left\{ \int_{\mathcal{X}} \inf_{\nu \in \mathcal{P}_p(\mathcal{Y})} \{C(x, \nu) - \int_{\mathcal{Y}} f(y) d\nu(y)\} d\mathbb{P}_0(x) + \int_{\mathcal{Y}} f(y) d\mathbb{P}_1(y) \right\} \geq$$

$$\int_{\mathcal{X}} \inf_{\nu \in \mathcal{P}_p(\mathcal{Y})} \{C(x, \nu) - \int_{\mathcal{Y}} f^*(y) d\nu(y)\} d\mathbb{P}_0(x) + \int_{\mathcal{Y}} f^*(y) d\mathbb{P}_1(y).$$

Now we use the fact that $\pi^*(\cdot|x)$ minimizes (8) for all $x \in \mathcal{X}$:

$$\int_{\mathcal{X}} \inf_{\nu \in \mathcal{P}_p(\mathcal{Y})} \{C(x, \nu) - \int f^*(y) d\nu(y)\} d\mathbb{P}_0(x) + \int_{\mathcal{Y}} f^*(y) d\mathbb{P}_1(y) =$$

$$= \int_{\mathcal{X}} \left\{ C(x, \pi^*(\cdot|x)) - \int_{\mathcal{Y}} f^*(y) d\pi^*(y|x) \right\} d\mathbb{P}_0(x) + \int_{\mathcal{Y}} f^*(y) d\mathbb{P}_1(y) =$$

$$\int_{\mathcal{X}} C(x, \pi^*(\cdot|x)) d\mathbb{P}_0(x) - \int_{\mathcal{X}} \int_{\mathcal{Y}} f^*(y) d\pi^*(y|x) \underbrace{d\mathbb{P}_0(x)}_{=d\pi_0^*(x)} + \int_{\mathcal{Y}} f^*(y) d\mathbb{P}_1(y) =$$

$$\int_{\mathcal{X}} C(x, \pi^*(\cdot|x)) d\mathbb{P}_0(x) - \int_{\mathcal{X} \times \mathcal{Y}} f^*(y) d\pi^*(x, y) + \int_{\mathcal{Y}} f^*(y) d\mathbb{P}_1(y) =$$

$$\int_{\mathcal{X}} C(x, \pi^*(\cdot|x)) d\mathbb{P}_0(x) - \int_{\mathcal{Y}} f^*(y) d\pi_1^*(y) + \int_{\mathcal{Y}} f^*(y) d\mathbb{P}_1(y) =$$

$$\int_{\mathcal{X}} C(x, \pi^*(\cdot|x)) d\mathbb{P}_0(x) + \underbrace{\int_{\mathcal{Y}} f^*(y) d(\mathbb{P}_1 - \pi_1^*)(y)}_{=0 \text{ since } \pi_1^* = \mathbb{P}_1} =$$

$$\int_{\mathcal{X}} C(x, \pi^*(\cdot|x)) d\mathbb{P}_0(x) = \mathbf{Cost}(\pi^*). \qquad (18)$$

We see that $\mathbf{Cost}(\pi^*)$ is not greater than the optimal $\mathbf{Cost}(\mathbb{P}_0, \mathbb{P}_1)$, i.e., $\pi^*$ is optimal. At the same time, from the derivations above, it directly follows that $f^*$ is an optimal potential. $\qquad \square$

*Proof of Theorem 3.2.* We are going to use our Theorem 3.1. First, we check that (13) holds for $\pi^*(\cdot|x)$ defined by (14). Analogously to [42, Theorem 1], for each $x \in \mathcal{X}$, we derive

$$\inf_{\nu \in \mathcal{P}_p(\mathcal{Y})} \{C_{c,\epsilon}(x, \nu) - \int_{\mathcal{Y}} f^*(y) d\nu(y)\} = \inf_{\nu \in \mathcal{P}_p(\mathcal{Y})} \underbrace{\left\{ \int_{\mathcal{Y}} \left[ c(x, y) - f^*(y) \right] d\nu(y) - \epsilon H(\nu) \right\}}_{\stackrel{\text{def}}{=} \mathcal{G}_x(\nu)}.$$

Minimizing $\mathcal{G}_x$, one should consider only $\nu \in \mathcal{P}_{p,ac}(\mathcal{Y}) \subset \mathcal{P}_p(\mathcal{Y})$. Indeed, for $\nu \notin \mathcal{P}_{p,ac}(\mathcal{Y})$, it holds that $\mathcal{G}_x(\nu^*) = +\infty$ since $c(x, y)$ is lower bounded and $-H(\nu) = +\infty$. We continue

$$\inf_{\nu \in \mathcal{P}_{p,ac}(\mathcal{Y})} \left\{ -\epsilon \int_{\mathcal{Y}} \log \exp \left( \frac{f^*(y) - c(x, y)}{\epsilon} \right) d\nu(y) + \overbrace{\epsilon \int_{\mathcal{Y}} \log \frac{d\nu(y)}{dy} d\nu(y)}^{=-H(\nu)} \right\} =$$

$$\inf_{\nu \in \mathcal{P}_{p,ac}(\mathcal{Y})} \left\{ -\epsilon \int_{\mathcal{Y}} \log \left( Z_x \cdot \frac{d\pi^*(y|x)}{dy} \right) d\nu(y) + \epsilon \int_{\mathcal{Y}} \log \frac{d\nu(y)}{dy} d\nu(y) \right\} =$$

$$-\epsilon \log Z_x + \inf_{\nu \in \mathcal{P}_{p,ac}(\mathcal{Y})} \left\{ -\epsilon \int_{\mathcal{Y}} \log \frac{d\pi^*(y|x)}{dy} d\nu(y) + \epsilon \int_{\mathcal{Y}} \log \frac{d\nu(y)}{dy} d\nu(y) \right\} =$$

$$-\epsilon \log Z_x + \inf_{\nu \in \mathcal{P}_{p,ac}} \epsilon \mathrm{KL}\left(\nu \| \pi^*(\cdot|x)\right). \qquad (19)$$

Since $\pi^*(\cdot|x) \in \mathcal{P}_{p,ac}(\mathcal{Y})$, by the assumption of the current Theorem, we conclude that it is the unique minimum of $\mathcal{G}_x(\nu)$ in $\mathcal{P}_{p,ac}(\mathcal{Y})$. Now to apply our Theorem 3.1, it remains to check that all its assumptions hold. We only have to check that $C_{c,\epsilon}$ given by (6) is lower bounded, jointly lower semi-continuous and convex in the second argument.

Analogously to (19), we derive

$$C_{c,\epsilon}(x,\nu) = \int_{\mathcal{Y}} c(x,y)d\nu(y) - \epsilon H(\nu) = \underbrace{-\epsilon \log M_x}_{\geq -\epsilon \log M} + \epsilon \underbrace{\mathrm{KL}\left(\nu \| \nu_x\right)}_{\geq 0} \geq -\epsilon \log M, \qquad (20)$$

where $\frac{d\nu_x(y)}{dy} \overset{\text{def}}{=} M_x^{-1} \exp\left(-\frac{c(x,y)}{\epsilon}\right)$. This provides a lower bound on the cost $C_{c,\epsilon}$. From the first equality in (20), we see that $C_{c,\epsilon}$ is jointly lower semi-continuous because the first term $\int_{\mathcal{Y}} c(x,y)d\nu(y)$ is jointly lower semi-continuous by the assumptions and the entropy term $-H(\nu)$ is lower semi-continuous in $\mathcal{P}_1(\mathcal{Y})$ [48, Ex. 45] and hence in $\mathcal{P}_p(\mathcal{Y})$ as well ($p \geq 1$). The last step is to note that $C_{c,\epsilon}(x,\nu)$ is convex in $\nu$ thanks to the convexity of $-H(\nu)$.

Finally, if $\int_{\mathcal{X}} C_{c,\epsilon}\left(x, \pi^*(\cdot|x)\right)d\mathbb{P}_0(x) < \infty$, then $-\int_{\mathcal{X}} H\left(\pi^*(\cdot|x)\right)$ is finite. Let $U$ be the subset of plans $\pi \subset \Pi(\mathbb{P}_0, \mathbb{P}_1)$ where $-\int_{\mathcal{X}} H\left(\pi(\cdot|x)\right)$ is finite. It is not empty since $\pi^* \in U$. At the same time, it is a convex set and functional $\pi \mapsto -\int_{\mathcal{X}} H\left(\pi(\cdot|x)\right)d\mathbb{P}_0(x)$ is **strictly** convex in $U$ thanks to the strict convexity of the (negative) entropy $\nu \mapsto -H(\nu)$ on the set of distributions where it is finite. Thus, $\pi \mapsto \int_{\mathcal{X}} C_{c,\epsilon}\left(x, \pi(\cdot|x)\right)d\mathbb{P}_0(x)$ is strictly convex in $U$ and $\pi^*$ is the unique minimum.

For completeness, we note that if $\int_{\mathcal{X}} C_{c,\epsilon}\left(x, \pi^*(\cdot|x)\right)d\mathbb{P}_0(x) = +\infty$, this situation is trivial, as the cost of every plan turns to be equal to $+\infty$. As a result, every plan is optimal.

$\square$

*Proof of Proposition 3.3.* Deriving the actual form of $\pi^*(\cdot|x)$ is an easy exercise. We substitute (15) into (14) and use the quadratic cost $c(x,y) = \frac{||y-x||^2}{2}$:

$$\frac{d\pi^*(y|x)}{dy} = \frac{1}{Z_x} \exp\left(\frac{f^*(y) - c(x,y)}{\epsilon}\right) =$$

$$\frac{1}{Z_x} \exp\left(\frac{\epsilon \log \sum_{n=1}^{N} w_n \mathcal{Q}(y|b_n, \epsilon^{-1}A_n) - \frac{||y-x||^2}{2}}{\epsilon}\right) =$$

$$\frac{1}{Z_x} \left(\sum_{n=1}^{N} w_n \mathcal{Q}(y|b_n, \epsilon^{-1}A_n)\right) \exp(-\frac{||y-x||^2}{2\epsilon}) =$$

$$\frac{1}{Z_x} \sum_{n=1}^{N} w_n \left(\mathcal{Q}(y|b_n, \epsilon^{-1}A_n) \exp(-\frac{||y-x||^2}{2\epsilon})\right) =$$

$$\frac{1}{Z_x} \sum_{n=1}^{N} w_n \left(\exp\left[-\frac{1}{2}(y-b_n)^T \frac{A_n}{\epsilon}(y-b_n)\right] \exp(-\frac{||y-x||^2}{2\epsilon})\right) =$$

$$\frac{1}{Z_x} \sum_{n=1}^{N} w_n \left(\exp\left[-\frac{1}{2}(y-b_n)^T \frac{A_n}{\epsilon}(y-b_n) - \frac{||y-x||^2}{2\epsilon}\right]\right) =$$

$$\frac{1}{Z_x} \sum_{n=1}^{N} w_n \left(\exp\left[-\frac{1}{2}(y-b_n)^T \frac{A_n}{\epsilon}(y-b_n) - \frac{1}{2}(y-x)^T \frac{I}{\epsilon}(y-x)\right]\right) =$$

$$\frac{1}{Z_x} \sum_{n=1}^{N} w_n \left(\exp\left[-\frac{1}{2}\{(y-b_n)^T \frac{A_n}{\epsilon}(y-b_n) + (y-x)^T \frac{I}{\epsilon}(y-x)\}\right]\right). \qquad (21)$$

Next, we prove that (we write just $\mu_n$ instead of $\mu_n(x)$ for simplicity):

$$(y-b_n)^T \frac{A_n}{\epsilon}(y-b_n) + (y-x)^T \frac{I}{\epsilon}(y-x) =$$

$$(y - \mu_n)^T \Sigma_n^{-1}(y - \mu_n) + (x - b_n)^T(\frac{I}{\epsilon} - \frac{\Sigma_n}{\epsilon^2})(x - b_n). \tag{22}$$

Indeed,

$$(y - b_n)^T \frac{A_n}{\epsilon}(y - b_n) + (y - x)^T \frac{I}{\epsilon}(y - x) =$$

$$y^T \frac{A_n}{\epsilon} y - 2b_n^T \frac{A_n}{\epsilon} y + b_n^T \frac{A_n}{\epsilon} b_n + y^T \frac{I}{\epsilon} y - 2x^T \frac{I}{\epsilon} y + x^T \frac{I}{\epsilon} x =$$

$$y^T \underbrace{(\frac{A_n + I}{\epsilon})}_{\Sigma_n^{-1}} y - 2(A_n b_n + x)^T \frac{I}{\epsilon} y + b_n^T \frac{A_n}{\epsilon} b_n + x^T \frac{I}{\epsilon} x =$$

$$y^T \Sigma_n^{-1} y - 2(A_n b_n + x)^T \frac{I}{\epsilon} y + b_n^T \frac{A_n}{\epsilon} b_n + x^T \frac{I}{\epsilon} x =$$

$$y^T \Sigma_n^{-1} y - 2 \underbrace{(A_n b_n + x)^T (A_n + I)^{-1}}_{\mu_n^T} \underbrace{\frac{(A_n + I)}{\epsilon}}_{\Sigma_n^{-1}} y + b_n^T \frac{A_n}{\epsilon} b_n + x^T \frac{I}{\epsilon} x =$$

$$y^T \Sigma_n^{-1} y - 2\mu_n^T \Sigma_n^{-1} y + b_n^T \frac{A_n}{\epsilon} b_n + x^T \frac{I}{\epsilon} x =$$

$$y^T \Sigma_n^{-1} y - 2\mu_n^T \Sigma_n^{-1} y + \mu_n^T \Sigma_n^{-1} \mu_n - \mu_n^T \Sigma_n^{-1} \mu_n + b_n^T \frac{A_n}{\epsilon} b_n + x^T \frac{I}{\epsilon} x =$$

$$(y - \mu_n)^T \Sigma_n^{-1}(y - \mu_n) - \mu_n^T \Sigma_n^{-1} \mu_n + b_n^T \frac{A_n}{\epsilon} b_n + x^T \frac{I}{\epsilon} x =$$

$$(y - \mu_n)^T \Sigma_n^{-1}(y - \mu_n) + b_n^T \frac{A_n}{\epsilon} b_n - \mu_n^T \Sigma_n^{-1} \mu_n + x^T \frac{I}{\epsilon} x =$$

$$(y - \mu_n)^T \Sigma_n^{-1}(y - \mu_n) + b_n^T \frac{A_n}{\epsilon} b_n - (A_n b_n + x)^T \frac{\Sigma_n}{\epsilon} \Sigma_n^{-1} \frac{\Sigma_n}{\epsilon}(A_n b_n + x) + x^T \frac{I}{\epsilon} x =$$

$$(y - \mu_n)^T \Sigma_n^{-1}(y - \mu_n) + b_n^T \frac{A_n}{\epsilon} b_n - (A_n b_n + x)^T \frac{\Sigma_n}{\epsilon^2}(A_n b_n + x) + x^T \frac{I}{\epsilon} x =$$

$$(y - \mu_n)^T \Sigma_n^{-1}(y - \mu_n) + b_n^T \frac{A_n}{\epsilon} b_n -$$

$$(A_n b_n)^T \frac{\Sigma_n}{\epsilon^2} A_n b_n - 2(A_n b_n)^T \frac{\Sigma_n}{\epsilon^2} x - x^T \frac{\Sigma_n}{\epsilon^2} x + x^T \frac{I}{\epsilon} x =$$

$$(y - \mu_n)^T \Sigma_n^{-1}(y - \mu_n) + b_n^T \frac{A_n}{\epsilon} b_n - (A_n b_n)^T \frac{\Sigma_n}{\epsilon^2} A_n b_n -$$

$$2(A_n b_n)^T \frac{\Sigma_n}{\epsilon^2} x + x^T(\frac{I}{\epsilon} - \frac{\Sigma_n}{\epsilon^2}) x =$$

$$(y - \mu_n)^T \Sigma_n^{-1}(y - \mu_n) + b_n^T \frac{A_n}{\epsilon} b_n - b_n^T \frac{A_n^T \Sigma_n A_n}{\epsilon^2} b_n -$$

$$2(A_n b_n)^T \frac{\Sigma_n}{\epsilon^2} x + x^T(\frac{I}{\epsilon} - \frac{\Sigma_n}{\epsilon^2}) x =$$

$$(y - \mu_n)^T \Sigma_n^{-1}(y - \mu_n) + b_n^T \frac{A_n - A_n^T \frac{\Sigma_n}{\epsilon} A_n}{\epsilon} b_n - 2(A_n b_n)^T \frac{\Sigma_n}{\epsilon^2} x + x^T(\frac{I}{\epsilon} - \frac{\Sigma_n}{\epsilon^2}) x =$$

$$(y - \mu_n)^T \Sigma_n^{-1}(y - \mu_n) + b_n^T \frac{A_n - A_n^T \frac{\Sigma_n}{\epsilon} A_n}{\epsilon} b_n - 2b_n^T \frac{A_n \Sigma_n}{\epsilon^2} x + x^T(\frac{I}{\epsilon} - \frac{\Sigma_n}{\epsilon^2}) x =$$

$$(y - \mu_n)^T \Sigma_n^{-1}(y - \mu_n) + b_n^T \frac{A_n - A_n^T \frac{\Sigma_n}{\epsilon} A_n}{\epsilon} b_n -$$

$$2b_n^T \frac{\overbrace{(A_n + I)}^{\epsilon \Sigma_n^{-1}} \Sigma_n - \Sigma_n}{\epsilon^2} x + x^T(\frac{I}{\epsilon} - \frac{\Sigma_n}{\epsilon^2}) x =$$

$$(y - \mu_n)^T \Sigma_n^{-1}(y - \mu_n) + b_n^T \frac{A_n - A_n^T \frac{\Sigma_n}{\epsilon} A_n}{\epsilon} b_n - 2b_n^T \frac{\epsilon I - \Sigma_n}{\epsilon^2} x + x^T(\frac{I}{\epsilon} - \frac{\Sigma_n}{\epsilon^2}) x =$$

$$(y - \mu_n)^T \Sigma_n^{-1}(y - \mu_n) + b_n^T \frac{A_n - A_n^T \frac{\Sigma_n}{\epsilon} A_n}{\epsilon} b_n - 2b_n^T(\frac{I}{\epsilon} - \frac{\Sigma_n}{\epsilon^2})x + x^T(\frac{I}{\epsilon} - \frac{\Sigma_n}{\epsilon^2})x =$$

$$(y - \mu_n)^T \Sigma_n^{-1}(y - \mu_n) + b_n^T \frac{A_n - A_n^T \frac{\Sigma_n}{\epsilon} A_n}{\epsilon} b_n -$$

$$\color{red}{b_n^T(\frac{I}{\epsilon} - \frac{\Sigma_n}{\epsilon^2})b_n + (x - b_n)^T(\frac{I}{\epsilon} - \frac{\Sigma_n}{\epsilon^2})(x - b_n)} =$$

$$(y - \mu_n)^T \Sigma_n^{-1}(y - \mu_n) + (x - b_n)^T(\frac{I}{\epsilon} - \frac{\Sigma_n}{\epsilon^2})(x - b_n) +$$

$$b_n^T \frac{A_n - A_n^T \frac{\Sigma_n}{\epsilon} A_n}{\epsilon} b_n - b_n^T(\frac{I}{\epsilon} - \frac{\Sigma_n}{\epsilon^2})b_n =$$

$$(y - \mu_n)^T \Sigma_n^{-1}(y - \mu_n) + (x - b_n)^T(\frac{I}{\epsilon} - \frac{\Sigma_n}{\epsilon^2})(x - b_n) +$$

$$\color{red}{b_n^T(\frac{A_n - A_n^T \frac{\Sigma_n}{\epsilon} A_n}{\epsilon} - \frac{I}{\epsilon} + \frac{\Sigma_n}{\epsilon^2})b_n} =$$

$$(y - \mu_n)^T \Sigma_n^{-1}(y - \mu_n) + (x - b_n)^T(\frac{I}{\epsilon} - \frac{\Sigma_n}{\epsilon^2})(x - b_n) +$$

$$\color{red}{b_n^T(\frac{A_n(I - \frac{\Sigma_n}{\epsilon} A_n)}{\epsilon} - \frac{I}{\epsilon} + \frac{\Sigma_n}{\epsilon^2})b_n} =$$

$$(y - \mu_n)^T \Sigma_n^{-1}(y - \mu_n) + (x - b_n)^T(\frac{I}{\epsilon} - \frac{\Sigma_n}{\epsilon^2})(x - b_n) +$$

$$\color{red}{b_n^T(\frac{A_n(I - \frac{\Sigma_n}{\epsilon}(\epsilon\Sigma_n^{-1} - I))}{\epsilon} - \frac{I}{\epsilon} + \frac{\Sigma_n}{\epsilon^2})b_n} =$$

$$(y - \mu_n)^T \Sigma_n^{-1}(y - \mu_n) + (x - b_n)^T(\frac{I}{\epsilon} - \frac{\Sigma_n}{\epsilon^2})(x - b_n) +$$

$$\color{red}{b_n^T(\frac{A_n(\frac{\Sigma_n}{\epsilon}) - I + \frac{\Sigma_n}{\epsilon}}{\epsilon})b_n} =$$

$$(y - \mu_n)^T \Sigma_n^{-1}(y - \mu_n) + (x - b_n)^T(\frac{I}{\epsilon} - \frac{\Sigma_n}{\epsilon^2})(x - b_n) +$$

$$\color{red}{b_n^T(\frac{(\epsilon\Sigma_n^{-1} - I)\frac{\Sigma_n}{\epsilon} - I + \frac{\Sigma_n}{\epsilon}}{\epsilon})b_n} =$$

$$(y - \mu_n)^T \Sigma_n^{-1}(y - \mu_n) + (x - b_n)^T(\frac{I}{\epsilon} - \frac{\Sigma_n}{\epsilon^2})(x - b_n) + \color{red}{b_n^T(\frac{I - \frac{\Sigma_n}{\epsilon} - I + \frac{\Sigma_n}{\epsilon}}{\epsilon})b_n} =$$

$$(y - \mu_n)^T \Sigma_n^{-1}(y - \mu_n) + (x - b_n)^T(\frac{I}{\epsilon} - \frac{\Sigma_n}{\epsilon^2})(x - b_n).$$

Next, we substitute (22) into (21)

$$\frac{1}{Z_x} \sum_{n=1}^{N} w_n \left( \exp\left[-\frac{1}{2}\{(y - b_n)^T \frac{A_n}{\epsilon}(y - b_n) + (y - x)^T \frac{I}{\epsilon}(y - x)\}\right]\right) =$$

$$\frac{1}{Z_x} \sum_{n=1}^{N} w_n \left( \exp\left[-\frac{1}{2}\{(y - \mu_n)^T \Sigma_n^{-1}(y - \mu_n) + (x - b_n)^T (\frac{I}{\epsilon} - \frac{\Sigma_n}{\epsilon^2})(x - b_n)\}\right]\right) =$$

$$\frac{1}{Z_x} \sum_{n=1}^{N} w_n \exp(-\frac{1}{2}(y - \mu_n)^T \Sigma_n^{-1}(y - \mu_n)) \exp(-\frac{1}{2}(x - b_n)^T (\frac{I}{\epsilon} - \frac{\Sigma_n}{\epsilon^2})(x - b_n)) =$$

$$\frac{1}{Z_x} \sum_{n=1}^{N} w_n (2\pi)^{\frac{D}{2}} \sqrt{\det(\Sigma_n)} \mathcal{N}(y|\mu_n, \Sigma_n) \mathcal{Q}(x|b_n, \frac{I}{\epsilon} - \frac{\Sigma_n}{\epsilon^2}) =$$

$$\frac{1}{Z_x} \sum_{n=1}^{N} \underbrace{w_n (2\pi)^{\frac{D}{2}} \sqrt{\det(\Sigma_n)} \mathcal{Q}(x|b_n, \frac{I}{\epsilon} - \frac{\Sigma_n}{\epsilon^2})}_{\widetilde{w}_n} \mathcal{N}(y|\mu_n, \Sigma_n)) =$$

$$\frac{1}{Z_x} \sum_{n=1}^{N} \widetilde{w}_n \mathcal{N}(y|\mu_n, \Sigma_n) = \frac{1}{\sum_{n=1}^{N} \widetilde{w}_n} \sum_{n=1}^{N} \widetilde{w}_n \mathcal{N}(y|\mu_n, \Sigma_n) =$$

$$\sum_{n=1}^{N} \frac{\widetilde{w}_n}{\sum_{n=1}^{N} \widetilde{w}_n} \mathcal{N}(y|\mu_n, \Sigma_n) = \sum_{n=1}^{N} \gamma_n \mathcal{N}(y|\mu_n, \Sigma_n).$$

which finishes the derivation of the expression for the density of $\pi^*(\cdot|x)$.

Now we prove that $\mathbb{P}_1 \stackrel{\text{def}}{=} \pi_1^* \in \mathcal{P}_2(\mathcal{Y})$. For each $x$, consider $\frac{d\pi^*(y|x)}{dy} = \sum_{n=1}^{N} \gamma_n \mathcal{N}(y|\mu_n(x), \Sigma_n)$. Its second moment is given by $\sum_{n=1}^{N} \gamma_n (\|\mu_n(x)\|^2 + \text{Tr} \, \Sigma_n)$. Note that

$$\|\mu_n(x)\| = \|(A_n + I)^{-1}(A_n b_n + x)\| \leq$$
$$\|(A_n + I)^{-1}\| \cdot \|A_n b_n + x\| \leq \|(A_n + I)^{-1}\| \cdot (\|A_n b_n\| + \|x\|),$$

where $\|\cdot\|$ applied to matrix means the operator norm. Hence, one may conclude that $\|\mu_n(x)\|^2$ is upper bounded by some quadratic polynomial of $\|x\|$, i.e., there exist constants $\alpha_n \in \mathbb{R}, \beta_n \in \mathbb{R}_+$ such that $\|\mu_n(x)\|^2 \leq \alpha_n + \beta_n \cdot \|x\|^2$. We derive

$$\int_{\mathcal{Y}} \|y\|^2 d\pi_1^*(y) = \int_{\mathcal{X}} \int_{\mathcal{Y}} \|y\|^2 d\pi^*(y|x) \underbrace{d\pi_0^*(x)}_{=d\mathbb{P}_0(x)} = \int_{\mathcal{X}} \sum_{n=1}^{N} \gamma_n (\|\mu_n(x)\|^2 + \text{Tr} \, \Sigma_n) d\mathbb{P}_0(x) \leq$$

$$\int_{\mathcal{X}} \sum_{n=1}^{N} \gamma_n (\alpha_n + \beta_n \|x\|^2 + \text{Tr} \, \Sigma_n) d\mathbb{P}_0(x) =$$

$$\sum_{n=1}^{N} \gamma_n (\alpha_n + \text{Tr} \, \Sigma_n) + \Big( \sum_{n=1}^{N} \beta_n \gamma_n \Big) \int_{\mathcal{X}} \|x\|^2 d\mathbb{P}_0(x) < \infty$$

since $\mathbb{P}_0 \in \mathcal{P}_2(\mathcal{X})$ by the assumption of the proposition.

It remains to prove that $\pi^*$ is the unique EOT plan. According to our Theorem 3.2, one only has to ensure that $\int_{\mathcal{X}} C_{c,\epsilon}(x, \pi^*(\cdot|x)) d\mathbb{P}_0(x) < \infty$. Just for completeness, we highlight that $\int_{\mathcal{X}} C_{c,\epsilon}(x, \pi^*(\cdot|x)) d\mathbb{P}_0(x)$ is *lower*-bounded since $C_{c,\epsilon}$ is lower bounded, see the proof of Theorem 3.2. Anyway, this is indifferent for us. We recall that $\pi^*$ is an optimal plan between $\mathbb{P}_0$ and $\mathbb{P}_1 = \pi_1^*$ and $f^*$ is an optimal potential by our construction. Thanks to the duality, we have

$$\int_{\mathcal{X}} C_{c,\epsilon}(x, \pi^*(\cdot|x)) d\mathbb{P}_0(x) = \int_{\mathcal{X}} (f^*)^{C_{c,\epsilon}}(x) d\mathbb{P}_0(x) + \int_{\mathcal{Y}} f^*(y) d\mathbb{P}_1(y) =$$

$$\int_{\mathcal{X}} \big[ -\epsilon \log Z_x \big] d\mathbb{P}_0(x) + \int_{\mathcal{Y}} f^*(y) d\mathbb{P}_1(y), \qquad (23)$$

where in transition to (23) we used our findings of line (19). Note that $\int_{\mathcal{Y}} f^*(y) d\mathbb{P}_1(y)$ is finite since $f^* \in \mathcal{C}_2(\mathcal{Y})$ is dominated by a quadratic polynomial, and we have already proved that $\mathbb{P}_1$ has finite second moment. It remains to upper bound the first term in (23). We note that

$$Z_x = \int_{\mathcal{Y}} \exp \left( \frac{f^*(y) - \frac{1}{2}\|x - y\|^2}{\epsilon} \right) dy = (\sqrt{2\pi\epsilon})^D \int_{\mathcal{Y}} \exp \left( \frac{f^*(y)}{\epsilon} \right) \mathcal{N}(y|x, \epsilon I) dy \geq$$

$$(\sqrt{2\pi\epsilon})^D \exp \left( \int_{\mathcal{Y}} \frac{f^*(y)}{\epsilon} \mathcal{N}(y|x, \epsilon I) dy \right) \geq (\sqrt{2\pi\epsilon})^D \exp \left( \int_{\mathcal{Y}} \frac{\beta + \alpha \|y\|^2}{\epsilon} \mathcal{N}(y|x, \epsilon I) dy \right) = \quad (24)$$

$$(\sqrt{2\pi\epsilon})^D \exp \left( \frac{\beta + \alpha(\|x\|^2 + \epsilon D)}{\epsilon} \right), \quad (25)$$

where in transition to line (24) we used the Jesnsen's inequality and $\alpha, \beta \in \mathbb{R}$ are some constants for which $f^*(\cdot) \geq \beta + \alpha \| \cdot \|^2$. They exist since $f^* \in \mathcal{C}_2(\mathcal{Y})$. Indeed, there exist $\tilde{\alpha}, \tilde{\beta} : |f^*(\cdot)| \leq \tilde{\beta} + \tilde{\alpha}\| \cdot \|^2 \Rightarrow f^*(\cdot) \geq -\tilde{\beta} - \tilde{\alpha}\| \cdot \|^2$, and we set $\alpha = -\tilde{\alpha}, \beta = -\tilde{\beta}$. In turn, line (25) uses the explicit formula for the second moment of $\mathcal{N}(y|x, \epsilon I)$. We use (25) to upper bound the first term in (23):

$$\int_{\mathcal{X}} \big[ -\epsilon \log Z_x \big] d\mathbb{P}_0(x) \leq \int_{\mathcal{X}} \big[ -\epsilon \log \left( \{ (\sqrt{2\pi\epsilon})^D \exp \left( \frac{\beta + \alpha \|x\|^2 + \alpha\epsilon D}{\epsilon} \right) \} \right) \big] d\mathbb{P}_0(x) =$$

$$-\frac{\epsilon D}{2}\log(2\pi\epsilon) - \beta - \alpha\epsilon D - \alpha\int_{\mathcal{X}}\|x\|^2 d\mathbb{P}_0(x).$$

It remains to note that the last value is finite, since $\mathbb{P}_0 \in \mathcal{P}_2(\mathcal{X})$ by the assumption. $\qquad\square$

*Proof of Corollary 3.4.* We note that $\frac{d\pi^*(y|x)}{dy} \propto \exp\left(\frac{f^*(y)-\frac{1}{2}\|x-y\|^2}{\epsilon}\right)$. Therefore,

$$\exp\left(\frac{f^*(y)}{\epsilon}\right) \propto \frac{d\pi^*(y|x)}{dy}\exp\left(\frac{1}{2\epsilon}\|x-y\|^2\right) \propto \frac{d\pi^*(y|x)}{dy}\cdot\left[\mathcal{N}(y|x,\epsilon I)\right]^{-1}. \qquad (26)$$

By comparing (26) with (11), we see that $\exp\left(\frac{f^*(y)}{\epsilon}\right)$ indeed coincides with the Schrödinger potential $\phi^*(y)$. Formula (12) for the optimal drift follows from [38, Proposition 4.1][3]. $\qquad\square$

*Proof of Corollary 3.5.* First, we prove that constructed $\mathbb{P}_1 \stackrel{\text{def}}{=} \pi_1^*$ actually has finite entropy. This is needed to ensure that the assumptions of [38, Proposition 4.1]. This proposition provides the formula for the optimal drift (12) via the Schrödinger potential. We write

$$0 \leq \mathrm{KL}\left(\pi_1^* \| \mathcal{N}(\cdot|0,I)\right) = -H(\pi_1^*) - \int_{\mathcal{Y}}\log\mathcal{N}(y|0,I)d\pi_1^*(y) =$$

$$-H(\pi_1^*) + \frac{D}{2}\log(2\pi) + \frac{1}{2}\int_{\mathcal{Y}}\|y\|^2 d\pi_1^*(y). \qquad (27)$$

From our Proposition 3.3 it follows that $\mathbb{P}_1 = \pi_1^*$ has finite second moment. Hence, the latter constant in (27) is finite. Therefore, $H(\pi_1^*)$ is upper bounded. To lower bound $H(\pi_1^*)$, recall that each $\pi^*(\cdot|x)$ is a mixture of $N$ Gaussians (Proposition 3.3) with ($x$-independent) covariances $\Sigma_n$. Thus, its density $\frac{d\pi^*(y|x)}{dy}$ is upper bounded by $\xi \stackrel{\text{def}}{=} \max_n\left[(2\pi)^{-D/2}\right](\det\Sigma_n)^{-1/2} > 0$ which also means that

$$\frac{d\pi_1^*(y)}{dy} = \int_{\mathcal{X}}\frac{d\pi^*(y|x)}{dy}d\pi_0^*(x) \leq \int_{\mathcal{X}}\xi d\pi_0^*(x) \leq \xi.$$

We conclude that

$$H(\pi_1^*) = -\int\log\frac{d\pi_1^*(y)}{dy}d\pi_1^*(y) \geq -\int\log\xi d\pi_1^*(y) = -\log\xi, \qquad (28)$$

i.e., $H(\pi_1^*)$ is lower-bounded as well.

Having in mind our previous Corollary, we just substitute $\exp\left(\frac{f^*(y)}{\epsilon}\right)$ of LSE (15) potential $f^*$ as the Schrödinger potential $\phi^*(y)$ to (12). We derive

$$v^*(x,t) = \epsilon\nabla\log\int_{\mathbb{R}^D}\mathcal{N}(y|x,(1-t)\epsilon I)\varphi^*(y)dy =$$

$$\epsilon\nabla\log\int_{\mathbb{R}^D}\mathcal{N}(y|x,(1-t)\epsilon I)\exp(\frac{f^*(y)}{\epsilon})dy =$$

$$\epsilon\nabla\log\int_{\mathbb{R}^D}\mathcal{N}(y|x,(1-t)\epsilon I)\exp(\frac{\epsilon\log\sum_{n=1}^N w_n\mathcal{Q}(y|b_n,\epsilon^{-1}A_n)}{\epsilon})dy =$$

$$\epsilon\nabla\log\sum_{n=1}^N w_n\int_{\mathbb{R}^D}\mathcal{N}(y|x,(1-t)\epsilon I)\mathcal{Q}(y|b_n,\epsilon^{-1}A_n))dy =$$

$$\epsilon\nabla\log\sum_{n=1}^N w_n\int_{\mathbb{R}^D}\left(2\pi\epsilon(1-t)\right)^{-\frac{D}{2}}\exp(-(y-x)^T\frac{I}{2\epsilon(1-t)}(y-x))\mathcal{Q}(y|b_n,\epsilon^{-1}A_n)dy =$$

---

[3]The authors of [38] consider SB with the *reversible* Wiener prior $R$, i.e., the standard Brownian motion starting at the Lebesgue measure. They deal with $\inf_{T\in\mathcal{F}(\mathbb{P}_0,\mathbb{P}_1)}\mathrm{KL}\left(T\|R\right)$ which matches (up to an additive constant) our formulation (9) for $\epsilon = 1$. Indeed, using the measure disintegration theorem, one can derive $\mathrm{KL}\left(T\|R\right) = -H(\mathbb{P}_0) + \mathrm{KL}\left(T\|W^\epsilon\right)$. For other $\epsilon > 0$, the analogous equivalence holds true.

$$\epsilon \nabla \log \sum_{n=1}^{N} w_n \int_{\mathbb{R}^D} \exp(-(y-x)^T \frac{I}{2\epsilon(1-t)}(y-x)) \mathcal{Q}(y|b_n, \epsilon^{-1}A_n) dy +$$

$$\underbrace{\epsilon \nabla \log \left( (2\pi\epsilon(1-t))^{-\frac{D}{2}} \right)}_{=0} =$$

$$\epsilon \nabla \log \sum_{n=1}^{N} w_n \int_{\mathbb{R}^D} \exp(-(y-x)^T \frac{I}{2\epsilon(1-t)}(y-x)) \exp(-(y-b_n)^T \frac{A_n}{2\epsilon}(y-b_n)) dy =$$

$$\epsilon \nabla \log \sum_{n=1}^{N} w_n \int_{\mathbb{R}^D} \exp\left( -\frac{1}{2(1-t)} \{ (y-x)^T \frac{I}{\epsilon}(y-x) + (y-b_n)^T \overbrace{\frac{(1-t)A_n}{\epsilon}}^{A_n^t} (y-b_n) \} \right) dy$$

Next, we use (22) but with $A_n^t$ instead of $A_n$ and $\Sigma_n^t$ instead of $\Sigma_n$. Also, we denote $\mu_n^t = (A_n^t + I)^{-1}(A_n^t b_n + x)$:

$$\epsilon \nabla \log \sum_{n=1}^{N} w_n \int_{\mathbb{R}^D} \exp\left( -\frac{1}{2(1-t)} \{ (y-x)^T \frac{I}{\epsilon}(y-x) + (y-b_n)^T \overbrace{\frac{(1-t)A_n}{\epsilon}}^{A_n^t} (y-b_n) \} \right) dy =$$

$$\epsilon \nabla \log \sum_{n=1}^{N} w_n \int_{\mathbb{R}^D} \exp\left( -\frac{1}{2(1-t)} \{ (y-\mu_n^t)^T \left( \Sigma_n^t \right)^{-1} (y-\mu_n^t) + \right.$$

$$\left. (x-b_n)^T (\frac{I}{\epsilon} - \frac{\Sigma_n^t}{\epsilon^2})(x-b_n) \} \right) dy =$$

$$\epsilon \nabla \log \sum_{n=1}^{N} \left\{ w_n \exp\left( -\frac{1}{2}(x-b_n)^T \frac{\epsilon I - \Sigma_n^t}{\epsilon^2(1-t)}(x-b_n) \right) \right.$$

$$\left. \int_{\mathbb{R}^D} \exp\left( -\frac{1}{2}(y-\mu_n^t)^T \frac{(\Sigma_n^t)^{-1}}{(1-t)}(y-\mu_n^t) \right) dy \right\} =$$

$$\epsilon \nabla \log \sum_{n=1}^{N} w_n \mathcal{Q}(x|b_n, \frac{\epsilon I - \Sigma_n^t}{\epsilon^2(1-t)}) \int_{\mathbb{R}^D} \exp\left( -\frac{1}{2}(y-\mu_n^t)^T \frac{(\Sigma_n^t)^{-1}}{(1-t)}(y-\mu_n^t) \right) dy =$$

$$\epsilon \nabla \log \sum_{n=1}^{N} w_n \mathcal{Q}(x|b_n, \frac{\epsilon I - \Sigma_n^t}{\epsilon^2(1-t)}) \int_{\mathbb{R}^D} (2\pi)^{\frac{D}{2}} \det((1-t)\Sigma_n^t)^{\frac{1}{2}} \mathcal{N}(y|\mu_n^t, (1-t)\Sigma_n^t) dy =$$

$$\epsilon \nabla \log \sum_{n=1}^{N} w_n \mathcal{Q}(x|b_n, \frac{\epsilon I - \Sigma_n^t}{\epsilon^2(1-t)})(2\pi(1-t))^{\frac{D}{2}} \det(\Sigma_n^t)^{\frac{1}{2}} \underbrace{\int_{\mathbb{R}^D} \mathcal{N}(y|\mu_n^t, (1-t)\Sigma_n^t) dy}_{=1} =$$

$$\epsilon \nabla \log \sum_{n=1}^{N} w_n \mathcal{Q}(x|b_n, \frac{\epsilon I - \Sigma_n^t}{\epsilon^2(1-t)})(2\pi(1-t))^{\frac{D}{2}} \det(\Sigma_n^t)^{\frac{1}{2}} =$$

$$\epsilon \nabla \log \sum_{n=1}^{N} w_n \mathcal{Q}(x|b_n, \frac{\epsilon I - \Sigma_n^t}{\epsilon^2(1-t)}) \det(\Sigma_n^t)^{\frac{1}{2}} + \underbrace{\epsilon \nabla \log \left( (2\pi(1-t))^{\frac{D}{2}} \right)}_{=0} =$$

$$\epsilon \nabla \log \sum_{n=1}^{N} w_n \sqrt{\det(\Sigma_n^t)} \mathcal{Q}(x|b_n, \frac{\epsilon I - \Sigma_n^t}{\epsilon^2(1-t)}),$$

which finishes the proof. $\qquad\square$

## B  Mixtures Benchmark Pairs: Details and Results

**Parameters for constructing benchmark pairs.** In our benchmark pairs, we choose all their hyperparameters manually to make sure the constructed distributions $\mathbb{P}_0, \mathbb{P}_1$ are visually pleasant and

distinguishable. As $\mathbb{P}_0$, we always use the centered Gaussian whose covariance matrix is $0.25I$. We use LSE function (15) with $N = 5$ for constructing the distribution $\mathbb{P}_1$. In each setup, all $A_n$ are the same and given in Table 3. We pick $w_n$ such that $\gamma_n = \frac{1}{5}\mathcal{N}(x|b_n, (\frac{1}{\epsilon}I - \frac{1}{\epsilon^2}\Sigma_n)^{-1})$. We sample $b_n$ randomly from a uniform distribution on a sphere with the radius $R = 5$.

|  | $D=2$ | $D=16$ | $D=64$ | $D=128$ |
|---|---|---|---|---|
| $\epsilon = 0.1$ | $\frac{1}{16}I$ | $\frac{1}{16}I$ | $\frac{1}{16}I$ | $\frac{1}{16}I$ |
| $\epsilon = 1$ | $\frac{1}{16}I$ | $\frac{1}{16}I$ | $\frac{1}{16}I$ | $\frac{1}{16}I$ |
| $\epsilon = 10$ | $\frac{9}{40}I$ | $\frac{1}{100}I$ | $\frac{1}{100}I$ | $\frac{1}{100}I$ |

Table 3: Matrices $A_n$ that we use to construct our mixtures benchmark pairs.

**Evaluation details.** For computing $\mathbb{BW}_2^2\text{-UVP}(\widehat{\pi}_1, \mathbb{P}_1)$, we use $10^5$ random samples from $\mathbb{P}_1$ and $10^5$ random samples from learned distribution $\widehat{\pi}_1$. For computing $c\mathbb{BW}_2^2\text{-UVP}(\widehat{\pi}, \pi^*)$, we use the hold-out test set containing 1000 samples $x \sim \mathbb{P}_0$. We compute the expectation and covariance matrices of $\pi^*(\cdot|x)$ analytically (Proposition 3.3) and we estimate the expectation and covariance matrix of $\widehat{\pi}(\cdot|x)$ by using $10^3$ samples. We present results of evaluation in Table 4 and Table 5.

We present an additional *trivial* baseline for the conditional metric $c\mathbb{BW}_2^2\text{-UVP}(\widehat{\pi}, \pi^*)$, which is given by the independent plan $\mathbb{P}_0 \times \mathbb{P}_1$. We compare other methods with this baseline in Table 5.

|  | $\epsilon = 0.1$ | | | | $\epsilon = 1$ | | | | $\epsilon = 10$ | | | |
|---|---|---|---|---|---|---|---|---|---|---|---|---|
|  | $D=2$ | $D=16$ | $D=64$ | $D=128$ | $D=2$ | $D=16$ | $D=64$ | $D=128$ | $D=2$ | $D=16$ | $D=64$ | $D=128$ |
| $\lfloor$LSOT$\rceil$ | - | - | - | - | - | - | - | - | - | - | - | - |
| $\lfloor$SCONES$\rceil$ | - | - | - | - | 1.06 | 4.24 | 6.67 | 11.54 | 1.11 | 2.98 | 1.33 | 7.89 |
| $\lfloor$NOT$\rceil$ | 0.016 | 0.63 | 1.53 | 2.62 | 0.08 | 1.13 | 1.62 | 2.62 | 0.225 | 2.603 | 1.872 | 6.12 |
| $\lfloor$EgNOT$\rceil$ | 0.09 | 0.31 | 0.88 | 0.22 | 0.46 | 0.3 | 0.85 | 0.12 | 0.077 | 0.02 | 0.15 | 0.23 |
| $\lfloor$ENOT$\rceil$ | 0.2 | 2.9 | 1.8 | 1.4 | 0.22 | 0.4 | 7.8 | 29 | 1.2 | 2 | 18.9 | 28 |
| $\lfloor$MLE-SB$\rceil$ | 0.01 | 0.14 | 0.97 | 2.08 | 0.005 | 0.09 | 0.56 | 1.46 | 0.01 | 1.02 | 6.65 | 23.4 |
| $\lfloor$DiffSB$\rceil$ | 2.88 | 2.81 | 153.22 | 232.67 | 0.87 | 0.99 | 1.12 | 1.56 | - | - | - | - |
| $\lfloor$FB-SDE-A$\rceil$ | 2.37 | 2.55 | 68.19 | 27.11 | 0.6 | 0.63 | 0.65 | 0.71 | - | - | - | - |
| $\lfloor$FB-SDE-J$\rceil$ | 0.03 | 0.05 | 0.25 | 2.96 | 0.07 | 0.13 | 1.52 | 0.48 | - | - | - | - |

Table 4: Comparisons of $\mathbb{BW}_2^2\text{-UVP} \downarrow$ (%) between the target $\mathbb{P}_1$ and learned marginal $\pi_1$. Colors indicate the metric value: $\mathbb{BW}_2^2\text{-UVP} \leq 0.5$, $\mathbb{BW}_2^2\text{-UVP} \in (0.5, 1]$, $\mathbb{BW}_2^2\text{-UVP} > 1.0$.

|  | $\epsilon = 0.1$ | | | | $\epsilon = 1$ | | | | $\epsilon = 10$ | | | |
|---|---|---|---|---|---|---|---|---|---|---|---|---|
|  | $D=2$ | $D=16$ | $D=64$ | $D=128$ | $D=2$ | $D=16$ | $D=64$ | $D=128$ | $D=2$ | $D=16$ | $D=64$ | $D=128$ |
| $\lfloor$LSOT$\rceil$ | - | - | - | - | - | - | - | - | - | - | - | - |
| $\lfloor$SCONES$\rceil$ | - | - | - | - | 34.88 | 71.34 | 59.12 | 136.44 | 32.9 | 50.84 | 60.44 | 52.11 |
| $\lfloor$NOT$\rceil$ | 1.94 | 13.67 | 11.74 | 11.4 | 4.77 | 23.27 | 41.75 | 26.56 | 2.86 | 4.57 | 3.41 | 6.56 |
| $\lfloor$EgNOT$\rceil$ | 129.8 | 75.2 | 60.4 | 43.2 | 80.4 | 74.4 | 63.8 | 53.2 | 4.14 | 2.64 | 2.36 | 1.31 |
| $\lfloor$ENOT$\rceil$ | 3.64 | 22 | 13.6 | 12.6 | 1.04 | 9.4 | 21.6 | 48 | 1.4 | 2.4 | 19.6 | 30 |
| $\lfloor$MLE-SB$\rceil$ | 4.57 | 16.12 | 16.1 | 17.81 | 4.13 | 9.08 | 18.05 | 15.226 | 1.61 | 1.27 | 3.9 | 12.9 |
| $\lfloor$DiffSB$\rceil$ | 73.54 | 59.7 | 1386.4 | 1683.6 | 33.76 | 70.86 | 53.42 | 156.46 | - | - | - | - |
| $\lfloor$FB-SDE-A$\rceil$ | 86.4 | 53.2 | 1156.82 | 1566.44 | 30.62 | 63.48 | 34.84 | 131.72 | - | - | - | - |
| $\lfloor$FB-SDE-J$\rceil$ | 51.34 | 89.16 | 119.32 | 173.96 | 29.34 | 69.2 | 155.14 | 177.52 | - | - | - | - |
| Independent | 166.0 | 152.0 | 126.0 | 110.0 | 86.0 | 80.0 | 72.0 | 60.0 | 4.2 | 2.52 | 2.26 | 2.4 |

Table 5: Comparisons of $c\mathbb{BW}_2^2\text{-UVP} \downarrow$ (%) between the optimal plan $\pi^*$ and the learned plan $\widehat{\pi}$.
**Colors** indicate the ratio of the metric to the *independent baseline* metric:
ratio $\leq 0.2$, ratio $\in (0.2, 0.5)$, ratio $> 0.5$.

**Colors for the Table 2.** To assign a color for the metric $\mathbb{BW}_2^2\text{-UVP}$ and $c\mathbb{BW}_2^2\text{-UVP}$ for each $\epsilon$ in the Table 2, we use the following rule: we assign the rank 1 if a method's metric for a given dimension $D$ has the color green, the rank 2 if a method's metric $\mathbb{BW}_2^2\text{-UVP}$ has the color orange and the rank 3 if a method's metric $\mathbb{BW}_2^2\text{-UVP}$ has the color red. To get the average rank, we take the mean of 4 ranks obtained for each dimension $D$ and round it (1.5 and 2.5 are rounded to 1 and 2 respectively).

**Extra qualitative results of EOT/SB solvers**. In Figure 5 and Figure 6, we present the additional qualitative comparison of solvers on our mixtures benchmark pairs in $D = 16$ with $\epsilon \in \{0.1, 10\}$. The figures are designed similarly to Figure 3 for $(D, \epsilon) = (16, 1)$ in the main text. Note that case $\epsilon = 10$ (Figure 6) is extremely challenging; only $\lfloor$**EgNOT**$\rceil$ provides more-or-less reasonable results.

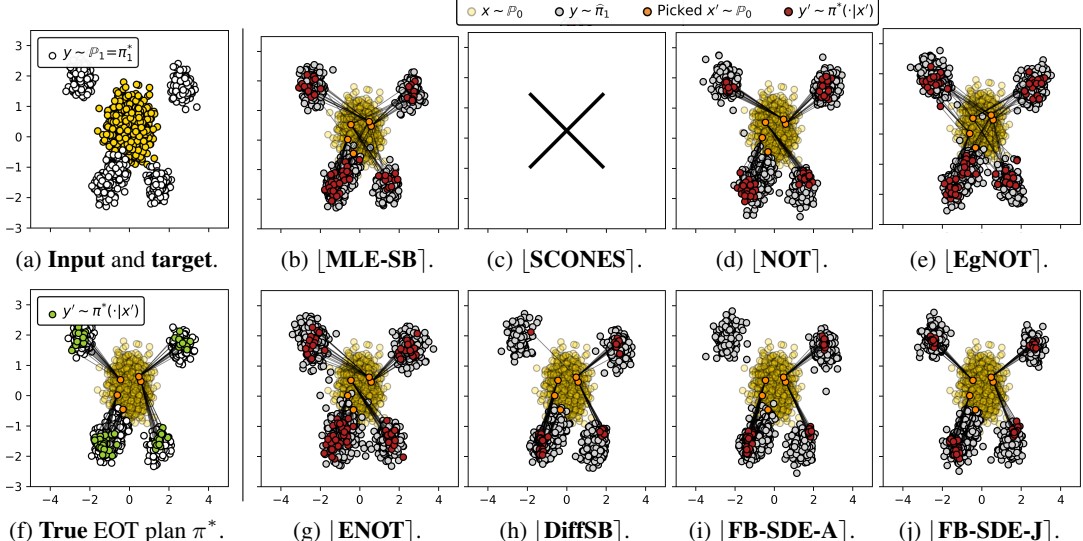

Figure 5: Qualitative results of EOT/SB solvers on our mixtures benchmark pair with $(D, \epsilon) = (16, 0.1)$. The distributions are visualized using 2 PCA components of target distr. $\mathbb{P}_1$.

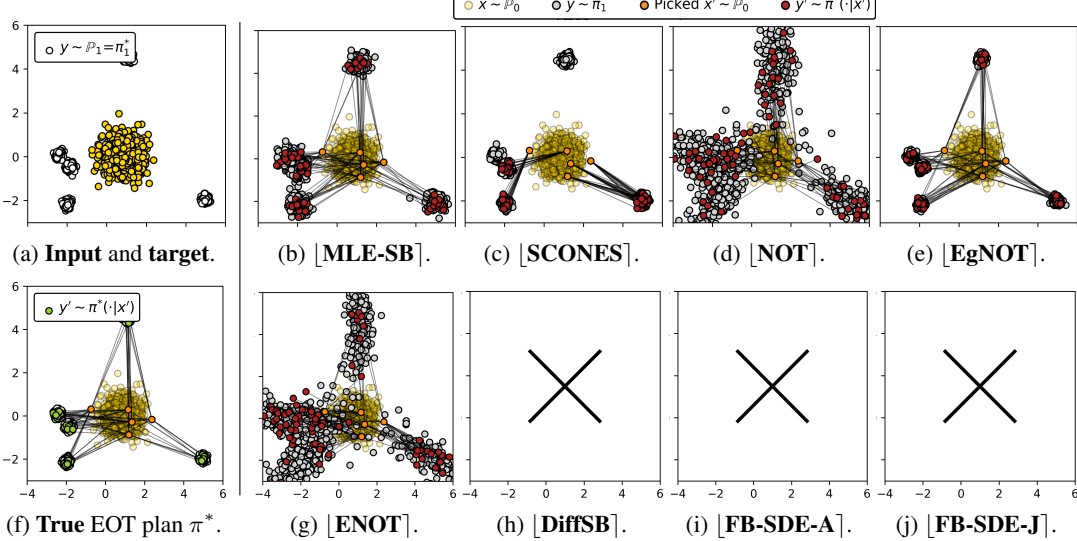

Figure 6: Qualitative results of EOT/SB solvers on our mixtures benchmark pair with $(D, \epsilon) = (16, 10)$. The distributions are visualized using 2 PCA components of target distr. $\mathbb{P}_1$.

**Computational complexity**. Sampling from $\mathbb{P}_0$ is lightspeed as it is just sampling a Normal noise. Sampling from $\mathbb{P}_1$ is also fast, as it is the Gaussian mixture (Proposition 3.3).

## C    Images Benchmark Pairs: Details and Results

**Parameters for constructing image benchmark pairs.** We fix $N = 100$ random samples from $\mathbb{P}_0$ for $b_n$ and choose all $A_n \equiv I$. We use $w_n$ such that $\gamma_n = \frac{1}{100} \mathcal{N}(x|b_n, (\frac{1}{\epsilon}I - \frac{1}{\epsilon^2}\Sigma_n)^{-1})$.

**GLOW details**. We use the code from the repository with the default parameters:

```
https://github.com/rosinality/glow-pytorch
```

After training, the latent variable $z$ is sampled from $N(0, \sigma^2 I)$ with $\sigma^2 = 0.49$ for image generation. That is, the image distribution $\mathbb{P}_0$ is produced by the mapping $z \sim N(0, \sigma^2 I)$ to the image space with the learned normalizing flow $G$, i.e., $\mathbb{P}_0 \stackrel{\text{def}}{=} G\sharp\mathcal{N}(\cdot|\sigma^2 I)$ in our construction.

**MCMC in the latent space of the normalizing flow.** We test EOT/SB solvers in $\mathbb{P}_1 \to \mathbb{P}_0$ direction, i.e., recovering $\pi^*(x|y)$ and generating clean samples $x$ from noised $y$. Unfortunately, the reverse conditional OT plans $\pi^*(x|y)$ are not as tractable as $\pi^*(y|x)$. However, we note that

$$\frac{d\pi^*(x|y)}{dy} \propto \frac{d\pi^*(y|x)}{dy}\frac{d\mathbb{P}_0(x)}{dx}, \tag{29}$$

i.e., the density of $\pi^*(\cdot|y)$ it known up to the normalizing constant. Recall that here $\mathbb{P}_0$ is constructed using the normalizing flow and $\pi^*(\cdot|x)$ is a Gaussian mixture (Proposition 3.3), i.e., we indeed know the values of both terms. Therefore, one may use the well-celebrated Langevin dynamics to sample from $\pi^*(y|x)$. Unfortunately, we found that such sampling in the image space is rather slow.

To overcome this issue, we employ the Langevin sampling in the latent space of the normalizing flow. It is possible since the normalizing flow is a bijection between the space of images and the latent space. We use the standard notation $z$ for the latent variable and $G : \mathbb{R}^D \to \mathbb{R}^D$ for the normalizing flow, i.e., $x = G(z) \sim \mathbb{P}_0$ for $z \sim p(z) \stackrel{\text{def}}{=} \mathcal{N}(z|0, \sigma^2 I)$. In this case, we have

$$\frac{d\pi^*(z|y)}{dz} = \frac{d\pi^*(x|y)}{dx}|\det J_{G^{-1}}(x)| \propto \frac{d\pi^*(y|x)}{dy}\frac{d\mathbb{P}_0(x)}{dx}|\det J_{G^{-1}}(x)| =$$

$$\frac{d\pi^*\big(y|G(z)\big)}{dx}\underbrace{\frac{d\mathbb{P}_0(x)}{dx}|\det J_{G^{-1}}(x)|}_{p(z)} = \frac{d\pi^*\big(y|G(z)\big)}{dy}p(z),$$

and we can derive the *score function* $\nabla_z \log \frac{d\pi^*(z|y)}{dz}$ which is needed for the Langevin dynamic as

$$\nabla_z \log \frac{d\pi^*(z|y)}{dz} = \nabla_z \frac{d\pi^*(y|G(z))}{dy} + \nabla_z \log p(z). \tag{30}$$

Hence, instead of doing non-trivial Langevin in the data space with $\nabla_x \frac{d\pi^*(x|y)}{dx}$, one may equivalently do the sampling in the latent space by using the score (30) and then get $x = G(z)$. We empirically found this approach works much better, presumably due to the fact that (30) is just the score of the Normal distribution which is slightly adjusted with the information coming from $\pi^*\big(y|G(z)\big)$.

For sampling, we employ the **Metropolis-adjusted Langevin algorithm** with the time steps $10^{-3}, 10^{-4}$ and $10^{-5}$ for $\epsilon = 10$, $\epsilon = 1$ and $\epsilon = 0.1$, respectively. It provides the theoretical guarantees that the constructed Markov chain $z_1, z_2, \ldots, \ldots$ converges to the distribution $\frac{d\pi^*(z|y)}{dz}$. For initializing the Markov chain, we sample a pair $(x, y) \sim \pi^*$ and use $z = G^{-1}(x)$ as the initial state for the Langevin sampling to get new samples from $\pi^*(\cdot|y)$. This trick allows for improving the stability of sampling and the convergence speed since it provides a good starting point. We use $N = 200$ steps for all the setups for the Metropolis-adjusted Langevin algorithm.

In Figures 7 and 8, we provide additional examples of the samples from the ground truth plan $\pi^*$.

**Metric 1.** For each $\epsilon = 0.1, 1, 10$ we prepare a test set with $10^4$ samples from $\mathbb{P}_0$. We use this set to calculate the FID [24] metric between the ground truth distribution $\mathbb{P}_0$ and the model's marginal distribution $\pi_1$ to estimate how well the model restores the target distribution. This allows to access the generative performance of solvers, i.e., the quality of generated images and matching the target distribution. However, *this metric does not assess the accuracy of the recovered EOT plan.*

**Metric 2.** For each $\epsilon = 0.1, 1, 10$, we prepare a test set containing 100 "noised" samples $y \sim \mathbb{P}_1$ and 5K samples $x \sim \pi^*(\cdot|y)$ for each "noised" sample $y$, i.e., 5K $\times$ 100 images for each $\epsilon$ in consideration. We propose to compute **conditional FID** to evaluate the difference between the conditional plans $\pi^*(\cdot|y)$ and $\widehat{\pi}(\cdot|y)$. That is, for each $y$ we compute FID between $\pi^*(\cdot|y)$ and $\widehat{\pi}(\cdot|y)$, and then average the result for all test $y$. Clearly, such an evaluation is approximately 100$\times$times more consuming than computing the base FID. However, *it allows us to fairly assess the quality of the recovered EOT solution, and we recommend this metric as the main for future EOT/SB studies.*

In Tables 6, 7, we present the evaluation results for $\lfloor$**ENOT**$\rceil$ [23]. We again emphasize that, to the best of our knowledge, there is no scalable *data→data* EOT/SB solver to compare against. Hence, we report the results as-is for future methods to be able to compare with them as the baseline.

**Computational complexity**. Sampling $x \sim \mathbb{P}_0$ is just applying the trained GLOW neural network to noise vectors $z \sim \mathcal{N}(\cdot|0, \sigma^2 I)$. Sampling $y \sim \mathbb{P}_1$ (or $y|x$) takes comparable time, as it is just extra

sampling from the Gaussian mixture with $x$-dependent parameters (Proposition 3.3). In turn, as we noted above, sampling $x|y$ requires using the Langevin dynamic and takes considerable time. To obtain 3 **test** sets of 5K samples $y \sim \pi^*(\cdot|x)$ per each of 100 samples $x \sim \mathbb{P}_0$, we employed 8×A100 GPUs. This generation of test datasets took approximately 1 week.

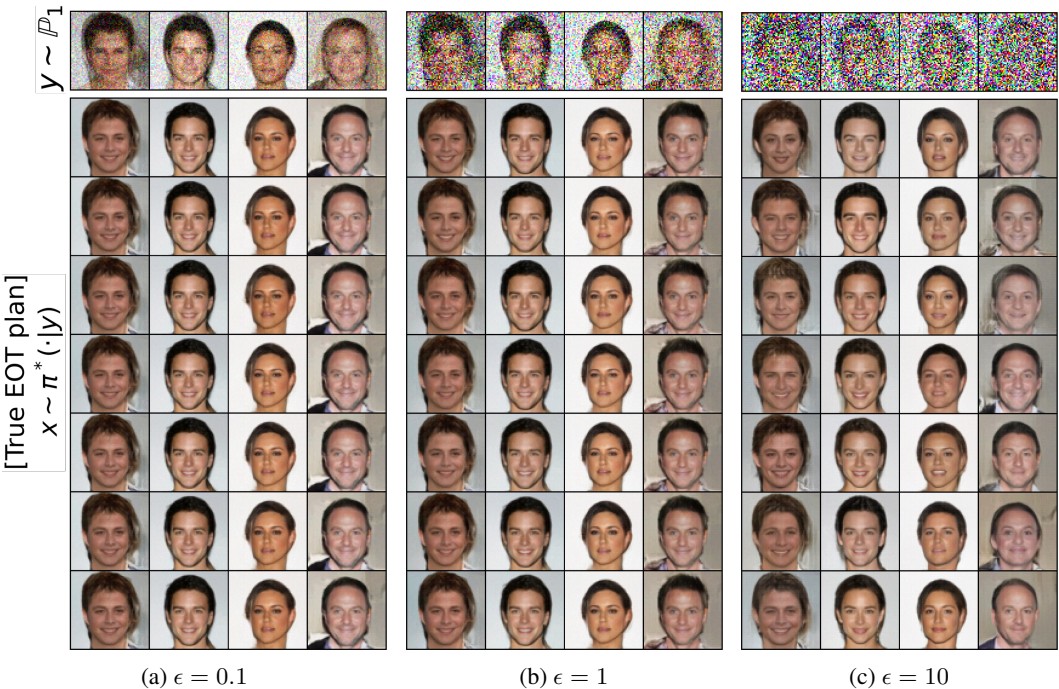

(a) $\epsilon = 0.1$          (b) $\epsilon = 1$          (c) $\epsilon = 10$

Figure 7: Ground truth samples $x \sim \pi^*(\cdot|y)$ on images benchmark pairs.

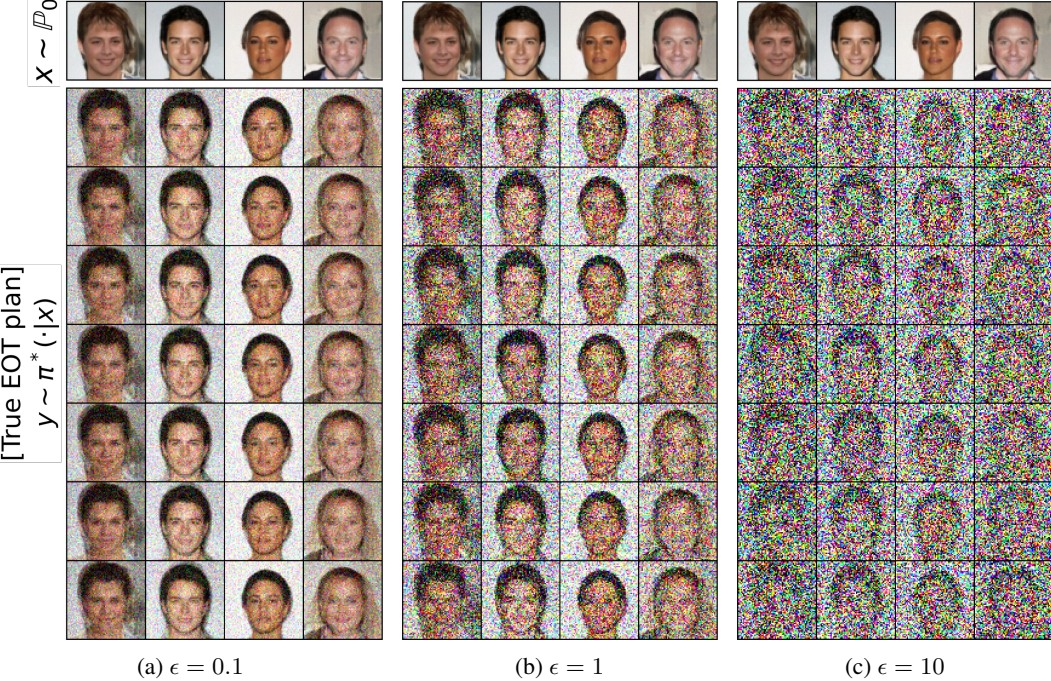

(a) $\epsilon = 0.1$          (b) $\epsilon = 1$          (c) $\epsilon = 10$

Figure 8: Ground truth samples $y \sim \pi^*(\cdot|x)$ on images benchmark pairs.

| $\epsilon$ | **0.1** | **1** | **10** |
|---|---|---|---|
| **FID** | 5.99 | 3.21 | 4.9 |

Table 6: Test FID of $\lfloor$**ENOT**$\rceil$ on our images benchmark pairs.

| $\epsilon$ | **0.1** | **1** | **10** |
|---|---|---|---|
| **cFID** | 40.5 | 19.8 | 14.47 |

Table 7: Test conditional FID of $\lfloor$**ENOT**$\rceil$ on our images benchmark pairs

## D  Details of EOT/SB Solvers

### D.1  Mixtures Benchmark Pairs

$\lfloor$**LSOT**$\rceil$ [49]. We use the part of the code of $\lfloor$**SCONES**$\rceil$ solver from the authors' repository

```
https://github.com/mdnls/scones-synthetic/blob/main/cpat.py
```

corresponding to learning dual OT potentials `blob/main/cpat.py` and the barycentric projection `blob/main/bproj.py` in the Gaussian case with configuration `blob/main/config.py`.

$\lfloor$**SCONES**$\rceil$ [14]. We use the aforementioned official code for training of dual OT potentials. We employ `sklearn.mixture.GaussianMixture` with 20 components to approximate the score of the target distribution. For the rest, we employ their configuration `blob/main/config.py` with batch size=1024 and the learning rate for Langevin sampling is $5 \cdot 10^{-4}$.

$\lfloor$**NOT**$\rceil$ This algorithm [35, Algorithm 1] is a generic algorithm for weak OT. It works for transport costs $C(x, \pi(\cdot|x))$ which are straightforward to estimate by using samples of $\pi(\cdot|x)$. Entropic cost $C_{c,\epsilon}$ (6) does not fit this requirement, as it is not easy to estimate entropy from samples. To do it, one has to know the density of $\pi(\cdot|x)$. Thus, the authors of $\lfloor$**NOT**$\rceil$ skipped EOT setting. We fill this gap and do a minor modification to their algorithm. As the base implementation, we use

| $lr$ (potential and transport map) | $T_{\text{steps}}$ | $\sigma_z$ |
|---|---|---|
| $1e-4$ | 99 | 1.0 |

Table 8: $\lfloor$**NOT**$\rceil$ training parameters for the mixture benchmark pairs experiment.

```
https://github.com/iamalexkorotin/NeuralOptimalTransport
```

Instead of the multi-layer perceptron generator, we take a conditional normalizing flow with RealNVP architecture with context-dependent latent normal distribution. This enables the access to the density of $\pi(\cdot|x)$ and allows applying $\lfloor$**NOT**$\rceil$ algorithm to EOT. Our reimplementation is available at

```
https://github.com/Penchekrak/FlowNOT
```

Due to the decreased expressivity of RealNVP compared to MLP from $\lfloor$**NOT**$\rceil$, we do more optimization steps for the transport map before updating potential as well as larger parameter count compared to the original solver implementation for a similar task. We use the same set of hyperparameters across all experiments with different $(\epsilon, D)$. The hyperparameters are summarized in Table 8.

$\lfloor$**EgNOT**$\rceil$ [42] We use the official code for $\lfloor$**EgNOT**$\rceil$ from

```
https://github.com/PetrMokrov/Energy-guided-Entropic-OT.
```

For our mixture benchmark pairs experiment, we adapt the author's setup for the Gaussian-to-Gaussian experiment from their original paper [42, §5.2]. In particular, we use the same architectures of neural networks, see [42, Appendix C.2], but change the hyper-parameters of [42, Algorithm 1], since the orig-

| $K$ | $K_{\text{test}}$ | $\sqrt{\eta}$ | $\sigma_0$ | $N$ |
|---|---|---|---|---|
| 500 | 1000 | 0.05 | 1.0 | 1024 |

Table 9: $\lfloor$**EgNOT**$\rceil$ training parameters for the mixture benchmark pairs experiment.

inal ones do not work properly when fitting Gaussian-to-Mixture. We hypothesize that the observed failure is due to the short-run nature of the energy-based training algorithm. We suppose that significantly increasing the number of Langevin steps $K$ used at the training stage may leverage the problem. The specific hyper-parameters of $\lfloor$**EgNOT**$\rceil$ algorithm are the same for all $(\epsilon, D)$ pairs and provided in Table 9.

We initialize the learning rate as $lr = 10^{-5}$ and decrease its value during the training. Similar to the original implementation of [42] we use a replay buffer but found that a high probability ($p = 0.95$) of samples reusage does not improve the quality and sometimes leads to unstable training. In turn, we choose $p = 0.5$. The reported numbers in Tables 4, 5 are gathered by launching the training process

for approximately 50K iterations and reporting the best-obtained metric. We understand that such an evaluation procedure is not ideal and does not provide statistically significant results. However, the qualitative results reported in Table 2 seem to show the behaviour of ⌊**EgNOT**⌉ solver on our benchmark setup and reveal the key properties of the approach.

⌊**ENOT**⌉ [23] We use the official code from



`https://github.com/ngushchin/EntropicNeuralOptimalTransport`



We use the same hyperparameters for this setup as the authors [23, Appendix E], except the number of discretization steps N, which we set to 200 as well as for other Schrödinger Bridge based methods. We also change the learning rate of the potential to $3 \cdot 10^{-4}$ for the setups with $\epsilon = 10$.

⌊**MLE-SB**⌉ [52]. We tested the official code from



`https://github.com/franciscovargas/GP_Sinkhorn`



Instead of Gaussian processes, we used a neural network as for ⌊**ENOT**⌉. We use $N = 200$ discretization steps as for other SB solvers, $5000$ IPF iterations, and $512$ samples from distributions $\mathbb{P}_0$ and $\mathbb{P}_1$ in each of them. We use the Adam optimizer with $lr = 10^{-4}$ for optimization.

⌊**DiffSB**⌉[15]. We utilize the official code from



`https://github.com/JTT94/diffusion_schrodinger_bridge`



with their configuration `blob/main/conf/dataset/2d.yaml` for toy problems. We increase the number of steps of dynamics to 200 and the number of steps of the IPF procedure for dimensions 16, 64 and 128 to 30, 40 and 60, respectively.

⌊**FB-SDE-J**⌉[9]. We utilize the official code from



`https://github.com/ghliu/SB-FBSDE`



with their configuration `blob/main/configs/default_checkerboard_config.py` for the checkerboard-to-noise toy experiment, changing the number of steps of dynamics from 100 to 200 steps. Since their hyper-parameters are developed for their 2-dimensional experiments, we increase the number of iterations for dimensions 16, 64 and 128 to 15 000.

⌊**FB-SDE-A**⌉ [9]. We also take the code from the same repository as above. We base our configuration on the authors' one (`blob/main/configs/default_moon_to_spiral_config.py`) for the moon-to-spiral experiment. As earlier, we increase the number of steps of dynamics up to 200. Also, we change the number of training epochs during one IPF procedure for dimensions 16, 64 and 128 to 2,4 and 8 correspondingly.

### D.2 Images Benchmark Pairs

⌊**ENOT**⌉ [23] As well as for the mixtures benchmark pairs, we use the official code from



`https://github.com/ngushchin/EntropicNeuralOptimalTransport`



We use the same hyperparameters for this setup as the authors [23, Appendix F] except the batch size which we set to 16 (`/blob/main/notebooks/Image_experiments.ipynb`).

## E   Additional Study of Hyperparameters of Solvers

To show that the default solvers parameters described in Appendix D are already a good choice, we additionally try different values of some of the most important hyperparameters. We consider each of the solvers except ⌊**LSOT**⌉ because it is anyway known to poorly perform due to the systematic bias in its solutions [31, 32]. For the evaluation, we consider the mixtures benchmark pair with $D = 64$ and $\epsilon = 1$ where most of the solvers perform reasonably well. In the tables below, we use "∗" to mark the hyperparameters that we use for comparisons in §4.1.

For ⌊**ENOT**⌉ solver, we consider the number of inner and outer problem iterations during the optimization and present the results in Table 10. The obtained results show that the performance increases slowly with increasing number of iterations of both types.

| Outer iters \ Inner iters | 1 | 5 | 10 | 20 |
|---|---|---|---|---|
| 100 | 131.1 | 130.3 | 74.5 | 129.3 |
| 1000 | 28.77 | 47.36 | 25.91 | 20.16 |
| 10000 | 24.46 | 37.36 | 23.07∗ | **18.03** |

Table 10: Comparison of cB$\mathbb{W}_2^2$-UVP ↓ (%) for ⌊**ENOT**⌋ on mixtures benchmark pairs for $D = 64$, $\epsilon = 1$ and different hyperparameters.

For IPF-based SB solvers ⌊**MLE-SB**⌋, ⌊**DiffSB**⌋, ⌊**FB-SDE-A**⌋ and ⌊**FB-SDE-J**⌋, we try different numbers of IPF iterations and the number of samples used in each iteration. We present the results in Tables 11, 12, 13, 14. All of the IPF-based solvers learn an inversion of a diffusion process at each IPF step but they differ in the way how this is done. The typical number of IPF steps used by each algorithm is affected by this difference. The performance increases slowly with the increase of the two hyperparameters considered, at the cost of a proportional increase in iterations or in the number of samples used.

| IPF iters \ Samples per iter | 64 | 128 | 256 | 512 |
|---|---|---|---|---|
| 100 | 23.45 | 24.50 | 16.64 | 14.23 |
| 1000 | 16.95 | 15.35 | 10.71 | 8.74 |
| 5000 | 11.55 | 11.24 | 12.96 | **8.41**∗ |

Table 11: Comparison of cB$\mathbb{W}_2^2$-UVP ↓ (%) for ⌊**MLE-SB**⌋ on mixtures benchmark pairs for $D = 64$, $\epsilon = 1$ and different hyperparameters.

| IPF iters \ Samples per iter | 64 | 256 | 512 | 1024 |
|---|---|---|---|---|
| 16 | 62.66 | 60.42 | 58.88 | 57.02 |
| 32 | 62.90 | 59.42 | 57.76∗ | 55.08 |
| 64 | 62.84 | 59.46 | 57.78 | **55.01** |

Table 12: Comparison of cB$\mathbb{W}_2^2$-UVP ↓ (%) for ⌊**DiffSB**⌋ on mixtures benchmark pairs for $D = 64$, $\epsilon = 1$ and different hyperparameters.

| IPF iters \ Samples per iter | 64 | 256 | 512 |
|---|---|---|---|
| 15000 | 173.16 | 163.04 | 160.5∗ |
| 30000 | 168.86 | 165.06 | **156.5** |

Table 13: Comparison of cB$\mathbb{W}_2^2$-UVP ↓ (%) for ⌊**FB-SDE-J**⌋ on mixtures benchmark pairs for $D = 64$, $\epsilon = 1$ and different hyperparameters.

| IPF iters \ Samples per iter | 64 | 256 | 512 | 1024 |
|---|---|---|---|---|
| 16 | 40.86 | 40.43 | 39.76 | 37.74 |
| 32 | 40.44 | 38.90 | 38.36∗ | 35.46 |
| 64 | 40.00 | 38.86 | 38.31 | **35.4** |

Table 14: Comparison of cB$\mathbb{W}_2^2$-UVP ↓ (%) for ⌊**FB-SDE-A**⌋ on mixtures benchmark pairs for $D = 64$, $\epsilon = 1$ and different hyperparameters.

For ⌊**SCONES**⌋ and ⌊**EgNOT**⌋ solvers, we consider the number of Langevin steps and the Langevin step size and present the results in Table 15 and Table 16. For ⌊**SCONES**⌋ the results obtained show that the performance increases slowly with increasing Langevin steps and decreasing Langevin step size. For ⌊**EgNOT**⌋ the trends are slightly different, since the optimal Langevin step size seems to be in the interval $[0.1, 0.2]$. Anyway, our selected parameters are reasonable ones because specifying an enormously large number of Langevin steps for these solvers is sort of impractical.

Finally, for ⌊**NOT**⌋ we consider the number of inner problem steps and the hidden size of the used neural network (conditional normalizing flow). We present results in Table 17.

| Langevin steps / Langevin step size | 64 | 256 | 512 | 1024 |
|---|---|---|---|---|
| $10^{-4}$ | 92.35 | 89.17 | 86.48 | **86.33**∗ |
| $10^{-3}$ | 93.51 | 90.41 | 88.22 | 87.74 |

Table 15: Comparison of cB$\mathbb{W}_2^2$-UVP $\downarrow$ (%) for $\lfloor$**SCONES**$\rceil$ on mixtures benchmark pairs for $D = 64$, $\epsilon = 1$ and different hyperparameters.

| Langevin steps / Langevin step size | 100 | 200 | 500 | 1000 |
|---|---|---|---|---|
| 0.01 | 70.9 | 70.98 | 72.9 | 68.13 |
| 0.02 | 71.31 | 67.14 | 69.11 | 69.02 |
| 0.05 | 68.78 | 68.59 | 63.73∗ | 56.84 |
| 0.1 | 64.52 | 57.45 | 52.35 | 51.9 |
| 0.2 | 58.22 | 60.08 | 58.93 | **41.31** |

Table 16: Comparison of cB$\mathbb{W}_2^2$-UVP $\downarrow$ (%) for $\lfloor$**EgNOT**$\rceil$ on mixtures benchmark pairs for $D = 64$, $\epsilon = 1$ and different hyperparameters.

| Hidden size / Inner steps | 64 | 128 | 192 | 256 | 320 | 384 | 448 | 512 |
|---|---|---|---|---|---|---|---|---|
| 1 | 93.14 | 167.05 | 149.52 | 189.0 | 89.1 | 161.66 | 176.43 | 175.67 |
| 5 | 82.64 | 86.09 | 82.18 | 190.04 | 147.31 | 105.46 | 103.5 | 150.76 |
| 10 | 163.47 | 146.68 | 53.26 | 137.47 | 100.84 | 171.65 | 115.84 | 126.96 |
| 100 | 18.68 | 21.4 | **14.64** | 18.08 | 16.66 | 20.64∗ | 18.71 | 15.15 |
| 200 | 61.99 | 52.74 | 58.63 | 53.89 | 52.44 | 55.3 | 55.02 | 54.75 |

Table 17: Comparison of cB$\mathbb{W}_2^2$-UVP $\downarrow$ (%) for $\lfloor$**NOT**$\rceil$ on mixtures benchmark pairs for $D = 64$, $\epsilon = 1$ and different hyperparameters.

**Discussion.** From the results it can be seen that for the most solvers' dependence on the considered hyperparameters is almost monotonic and the hyperparameters chosen for the solver comparison on the mixtures setup are in the region where the metric growth is almost saturated.

## F Qualitative Evaluation of the Drift Learned with SB methods

Our benchmark primarily aimed at quantifying the recovered **conditional EOT plan** $\widehat{\pi}(\cdot|x)$. Thanks to our Proposition 3.5, our benchmark provides not only the ground truth conditional EOT plan $\pi^*(\cdot|x)$, but the **optimal SB drift** $v^*(x, t)$ as well. This means that for SB solvers we may additionally compare their recovered SB drift $\widehat{v}$ with the ground truth drift $v^*$. Here we do this for $\lfloor$**MLE-SB**$\rceil$, $\lfloor$**DiffSB**$\rceil$, $\lfloor$**ENOT**$\rceil$, $\lfloor$**FB-SDE-A**$\rceil$, $\lfloor$**FB-SDE-J**$\rceil$ solvers by using our mixtures pairs.

METRICS. Recall that $T_{v^*}$ is the Schrödinger bridge (10) and let $T_{\widehat{v}}$ denote the learned process:

$$dX_t = \widehat{v}(x, t)dt + \sqrt{\epsilon}dW_t, \qquad X_0 \sim \mathbb{P}_0.$$

Both $T_{v^*}$ and $T_{\widehat{v}}$ are diffusion processes which start at distribution $\mathbb{P}_0$ at $t = 0$ and have fixed volatility $\epsilon$. Their respective drifts are $v^*$ and $\widehat{v}$. For each time $t \in [0, 1]$, consider

$$\mathcal{L}_{\text{fwd}}^2[t] \stackrel{\text{def}}{=} \mathbb{E}_{T_{v^*}}\|v^*(X_t, t) - \widehat{v}(X_t, t)\|^2, \tag{31}$$

$$\mathcal{L}_{\text{rev}}^2[t] \stackrel{\text{def}}{=} \mathbb{E}_{T_{\widehat{v}}}\|v^*(X_t, t) - \widehat{v}(X_t, t)\|^2. \tag{32}$$

which are the expected squared differences between the ground truth $v^*$ and learned $\widehat{v}$ drifts at the time $t$. In (31), the expectation is w.r.t. $X_t$ coming from the true SB trajectories of $T_{v^*}$, while in (32) – w.r.t. the learned trajectories from $T_{\widehat{v}}$. Reporting this metric for all the time steps, all the mixtures pairs and solvers would be an overkill. In what follows, we use this metric for quantitative analysis.

**First**, for $D = 16$ and $\epsilon \in \{0.1, 10\}$, we plot these metrics (as a function of time $t$). The results for all the solvers are shown in Figure 9. **Second**, we provide Table 19 where for $D \in \{2, 16, 64, 128\}$ and $\epsilon \in \{0.1, 10\}$ report $\mathcal{L}^2$ metrics averaged over $t \in [0, 1]$. Namely, we report

$$\text{KL}\left(T_{v^*}\|T_{\widehat{v}}\right) \stackrel{\text{def}}{=} \frac{1}{2\epsilon}\int_0^1 \mathcal{L}_{\text{fwd}}^2[t]dt \quad \text{and} \quad \text{RKL}\left(T_{v^*}\|T_{\widehat{v}}\right) \stackrel{\text{def}}{=} \frac{1}{2\epsilon}\int_0^1 \mathcal{L}_{\text{rev}}^2[t]dt. \tag{33}$$

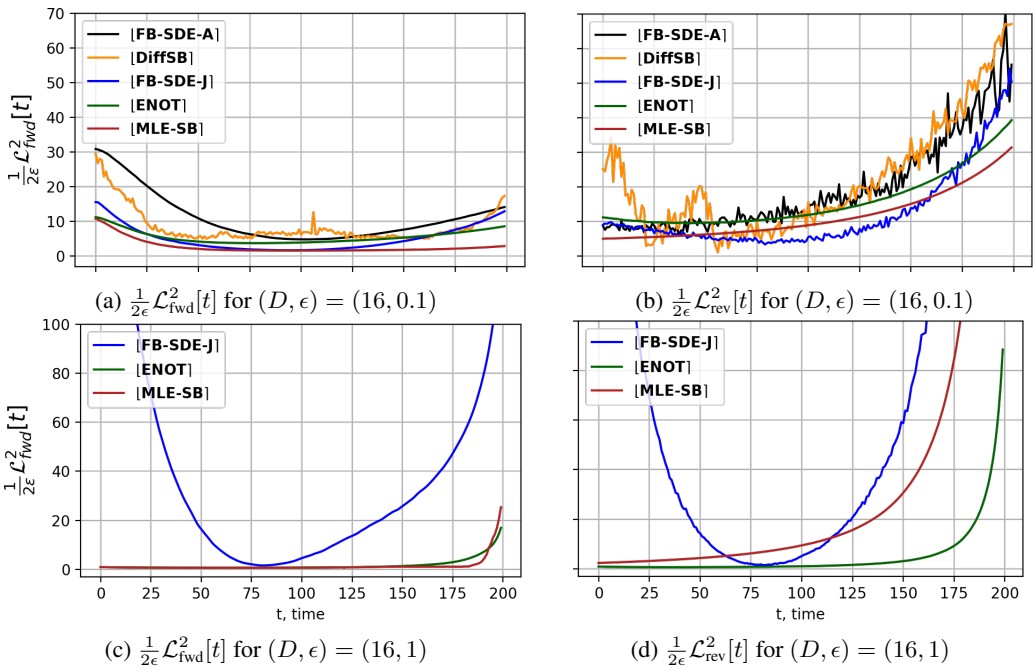

Figure 9: $\mathcal{L}^2$ metrics between the ground truth drift $v^*$ and the drift $\widehat{v}$ learned by SB solvers.

We write "KL" and "RKL" not by an accident. Thanks to the well-celebrated Girsanov's theorem, these are indeed the **forward and reverse KL divergences** between processes $T_{v^*}$ and $T_{\widehat{v}}$.

In all the SB solvers, we consider 200 time discretization steps $t = \{\frac{1}{200}, \frac{2}{200}, \dots 1\}$ for their training. During testing, we evaluate $\mathcal{L}^2$ metrics (31) and (32) on the same time steps. To estimate (31) and (32), we use $10^5$ samples $X_t$ which are taken from random trajectories of processes $T_{v^*}$ and $T_{\widehat{v}}$. These trajectories are simulated via the standard Euler–Maruyama method.

|  | $\epsilon = 0.1$ | | | | $\epsilon = 1$ | | | |
|---|---|---|---|---|---|---|---|---|
|  | $D=2$ | $D=16$ | $D=64$ | $D=128$ | $D=2$ | $D=16$ | $D=64$ | $D=128$ |
| ⌊**ENOT**⌉ | 0.61 | 5.49 | 6.59 | 10.36 | 0.86 | 1.64 | 11.43 | 37.53 |
| ⌊**DiffSB**⌉ | 6.96 | 12.89 | - | - | 12.28 | >1000 | >1000 | >1000 |
| ⌊**FB-SDE-A**⌉ | 6.9 | 11.08 | - | - | 10.59 | >1000 | >1000 | >1000 |
| ⌊**FB-SDE-J**⌉ | 3.02 | 5.02 | 9.60 | 28.85 | 18.79 | 44.79 | 629.28 | >1000 |
| ⌊**MLE-SB**⌉ | 0.62 | 2.63 | 4.76 | 7.86 | 0.96 | 1.86 | 9.66 | 34.95 |

Table 18: Forward KL between the ground truth SB process $T_{v^*}$ and the process $T_{\widehat{v}}$ learned with SB solvers on our mixtures benchmark pairs.

|  | $\epsilon = 0.1$ | | | | $\epsilon = 1$ | | | |
|---|---|---|---|---|---|---|---|---|
|  | $D=2$ | $D=16$ | $D=64$ | $D=128$ | $D=2$ | $D=16$ | $D=64$ | $D=128$ |
| ⌊**ENOT**⌉ | 72.86 | 78.98 | 135.29 | 221.26 | 18.40 | 49.65 | 177.02 | 348.05 |
| ⌊**DiffSB**⌉ | 11.85 | 21.16 | - | - | 121.43 | >1000 | >1000 | >1000 |
| ⌊**FB-SDE-A**⌉ | 12.29 | 19.40 | - | - | 100.22 | >1000 | >1000 | >1000 |
| ⌊**FB-SDE-J**⌉ | 8.03 | 12.11 | 17.16 | 49.32 | 64.37 | 123.68 | >1000 | >1000 |
| ⌊**MLE-SB**⌉ | 18.03 | 28.24 | 163.34 | 254.16 | 22.80 | 86.07 | 296.97 | 636.27 |

Table 19: Reverse KL between the ground truth SB process $T_{v^*}$ and the process $T_{\widehat{v}}$ learned with SB solvers on our mixtures benchmark pairs.

DISCUSSION. Interestingly, we see that the forward KL divergence shows a smoother behaviour than the RKL for almost all SB solvers. According to our evaluation, the ⌊**MLE-SB**⌉ and ⌊**ENOT**⌉ solvers mostly beat every other solver in the forward KL metric. At the same time, the RKL metric of ⌊**ENOT**⌉ is surprisingly the worst. While we make all these observations, we do not know how

to explain them. We hope that the question of the interpretation of the KL and RKL values will be addressed in future SB studies.

# G    Potential Societal Impact

Our proposed approach deals with generative models based on Entropic Optimal Transport and Schrödinger Bridge principles. Such models form and emergent subarea in the field of machine learning research and could be used for various purposes in the industry including image manipulation, artificial content rendering, graphical design, etc. Our benchmark is a step towards improving the reliability, robustness and transparency of these models. One potential negative of our work is that improving generative models may lead to transforming some jobs in the industry.

# H    Building Benchmarks from Real Data

In this section, we present a simple heuristic recipe to build benchmark pairs similar to some given real-world data. To illustrate the recipe, we consider toy 2D data example and several single-cell datasets [36, 8]. Code and data for the experiments in this section can be found in the `benchmark_construction_examplesdata` folder of our repository.

## H.1    Recipe for Building Benchmark Pairs form Data.

For constructing distribution pairs similar to some given data, we consider a pair of original and target datasets obtained from the true distributions $\mathbb{P}_0$ and $\mathbb{P}_1$, respectively. We heuristically initialize the LSE potential (15) $f^*(y) = \epsilon \log \sum_{n=1}^{N} w_n \mathcal{Q}(y|b_n, \epsilon^{-1}A_n)$ with $b_n$ as cluster centers obtained from the K-means clustering algorithm applied to the target data from $\mathbb{P}_1$. The weights $w_n$ are chosen to be $1/N$ and matrices $A_n = \lambda I$ are diagonal where $\lambda$ is a manually-chosen parameter (shared between all $A_n$). For any $x$ the conditional plan $\pi^*(\cdot|x)$ for LSE potential $f^*$ is just a Gaussian mixture and the mean of each its component is largely determined by $b_n$ (Proposition 3.3). We empirically found that the resulting constructed distribution $d\widehat{\mathbb{P}}_1(y) = d\pi_1^*(y) = \int d\pi^*(y|x)d\mathbb{P}_0(x)$ from $\mathbb{P}_0$ resembles the Gaussian mixture approximation of the target dataset if one managed to find proper value of $\lambda$.

In the rest of this section, we use the described recipe to construct benchmark pairs from data to show that the LSE parameterization of the potential provides a wide class of EOT/SB solutions and even allows constructing a benchmark similar to real data.

## H.2    Benchmark Pairs for 2D data.

Code and data for the experiment described in this section can be found in the folder `benchmark_construction_examples/2d_data` of our repository.

To begin with, we present the results of constructing a benchmark pair from 2D data. We consider a Gaussian distribution $\mathbb{P}_0$ as the source distribution and two moons $\mathbb{P}_1$ as the target distribution. We aim to use the previously described recipe §H.1 to find parameters of the LSE potential to construct an EOT solution between $\mathbb{P}_0$ and an approximation of $\mathbb{P}_1$ denoted as $\widehat{\mathbb{P}}_1$. Here we consider EOT with $\epsilon = 0.05$, use $N = 100$ for LSE potentials, and choose $\lambda = 50$. The result is in Figure 10.

As seen from the figure, the constructed target benchmark distribution $\widehat{\mathbb{P}}_1$ is similar to the target distribution $\mathbb{P}_1$. In turn, the EOT plan maps $x \sim \mathbb{P}_0$ to the close regions of the target distribution.

## H.3    Single-cell RNA Data

Code and data for the experiment described in this section can be found in the folder `benchmark_construction_examples/single_cell_rna` of our repository.

We consider the same setup as in [36, §5.2]. We use their data from the supplementary materials.[4] The provided data displays the progression of human embryonic stem cells as they differentiate from embryoid bodies into a range of cell types, such as mesoderm, endoderm, neuroectoderm, and neural

---

[4] `https://openreview.net/forum?id=d3QNWD_pcFv`

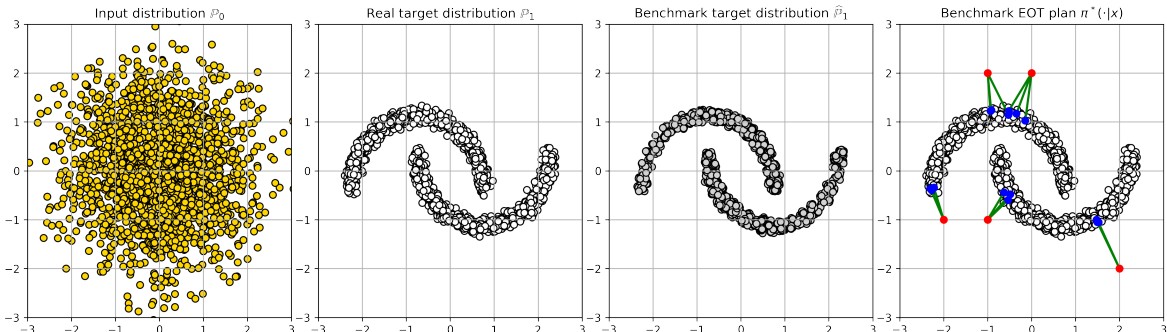

Figure 10: *Gaussian → Two Moons* benchmark pair.

crest, throughout a span of 27 days. The cell samples (approximately 2000 ones per each time period) were gathered at five distinct intervals ($t_0$: day 0 to 3, $t_1$: day 6 to 9, $t_2$: day 12 to 15, $t_3$: day 18 to 21, $t_4$: day 24 to 27). These collected cells were evaluated via scRNAseq, subjected to quality control filtering, and then projected onto a 5-dimensional feature space utilizing principal component analysis (PCA).

To construct the benchmark pair using the LSE potential, we consider $N = 250$, $\epsilon = 100$ and $\lambda = 100$ and employ the train data at times $t_0$ and $t_4$. Then we use the constructed benchmark plan $\pi^*(\cdot|x)$ to map source data at time $t_0$ to the data at time $t_4$ and obtain benchmark target distribution samples $\widehat{\mathbb{P}}_1$. Finally, we fit TSNE [51] to the combined dataset of samples from $\mathbb{P}_1$ and $\widehat{\mathbb{P}}_1$ and then plot their projections in Figure 11. The resulting plots are very similar, confirming that the constructed benchmark target data resembles the considered single-cell target data.

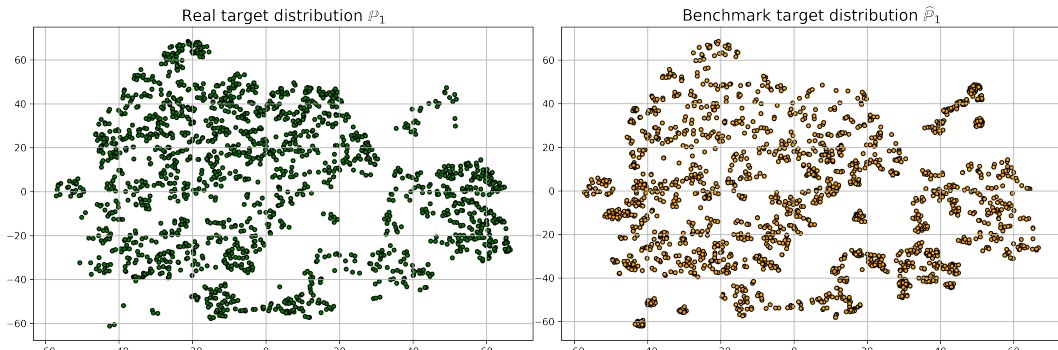

Figure 11: TSNE visualization of Single-cell RNA target data and our constructed target data.

## H.4 Single-cell Drugs Data

Code and data for the experiment described in this section can be found in the folder `benchmark_construction_examples/single_cell_drugs` of our repository.

| Method | scGen | cAE | CellOT [8] | EOT Benchmark (ours) |
|--------|-------|-----|-----------|----------------------|
| MMD↓ | 0.0241 | 0.0074 | 0.0013 | 0.0036 |

Table 20: MMD↓ distances (on the test data) between the observed perturbed cells $\mathbb{P}_1$ and predicted responses from control cells $\widehat{\mathbb{P}}_1$.

In [8], the authors consider the problem of predicting single-cell drug responses for drugs with different molecular effects, using melanoma cell lines profiled by 4i technology (single-cell technology). Utilizing a blend of two melanoma tumor cell lines at a 1:1 ratio, a total of 21,650 cells were imaged. Within this dataset, 11,526 cells existed in the untreated control state, 2,364 received Erlotinib treatment, 2,650 underwent Imatinib treatment, 2,683 were subjected to Trametinib treatment, and

2,417 were treated with a combination of Trametinib and Erlotinib. After preprocessing, each cell is described by 78 features. The train-test split with each drug is 80:20.

In this example, we consider cell data before treatment ($\mathbb{P}_0$) and after treatment with Erlotninib ($\mathbb{P}_1$). For the construction of the benchmark pair using an LSE potential, we consider $N = 250$, $\epsilon = 1$ and $\lambda = 20$. As with the single cell RNA data §H.3, we fit the TSNE [51] on a combined dataset of samples from $\mathbb{P}_1$ and $\widehat{\mathbb{P}}_1$ and then plot their projections in Figure 11. As seen from the visualizations, the TSNE projections of the real data and the mapped data are similar.

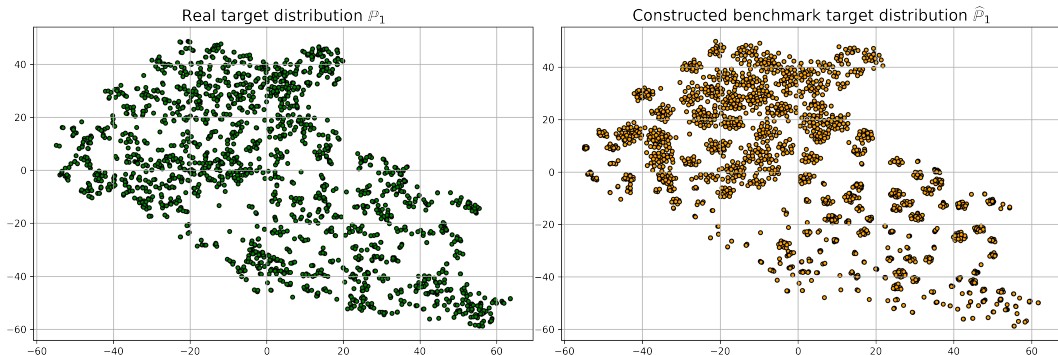

Figure 12: TSNE visualization of Single-cell Drugs target data and our constructed target data.

In addition, we quantitatively evaluate on the test data how well the constructed target distribution $\widehat{P}_1$ matches the true data distribution $\mathbb{P}_1$. We employ the same MMD metric as the authors and present the results in Table 20. The data for the baselines scGen, cAE and the authors' method CellOT are taken from [8]. As one can see, our approach is even better than two of the baselines considered.

