# OpenReview forum: "Building the Bridge of Schrödinger: A Continuous Entropic Optimal Transport Benchmark"
_NeurIPS.cc/2023/Track/Datasets_and_Benchmarks — NeurIPS 2023 Datasets and Benchmarks Poster_

### Official Review · Reviewer_p1pQ · 2023-07-20
**Review of "Building the Bridge of Schrödinger: A Continuous Entropic Optimal Transport Benchmark": Good and sound work. However, this proposed data construction method is limited.**

**Rating:** 6
**Confidence:** 4
**Clarity:** This paper is well written.

**Strengths:**

The topic is of great importance to the research community.
The idea is novel and the proofs seem to be correct.
The upside is the constructed pairs of probability distribution (i.e., P0 and P1) have analytically known SB, which may be beneficial for researchers to further develop and evaluate their algorithms.
The writing is good. The codebase is well-organized and easy to follow.

**Additional Feedback:**

The motivation of this paper is clear and strong.  This is a promising direction.
Providing a good benchmark for this community is necessary for further development.
More consideration about generality is needed to further improve this work.
Besides, it may be better to propose a dataset or a method to construct data that is more related to some real applications.

**Correctness:**

The claims are correct.
The benchmark data is constructed with sound math work.
The evaluation method is reasonable.

**Documentation:**

The reproducibility is well supported.

**Ethics:**

No.
No ethics issues are involved.

**Limitations:**

The authors have adequately addressed the limitations.

**Opportunities For Improvement:**

The potential function $f^*$ is chosen to be a weighted log-sum-exp of quadratic functions. It is clear that this choice could provide analytical EOT/SB solutions. However, this choice may conflict with generality and representativeness, which are important aspects, especially for a benchmark.

The authors have tested existing neural EOT/SB solvers. However, the experimental results may not be convincing enough due to the lack of tuning hyperparameters (I think the authors are quite aware of this point).

**Relation To Prior Work:**

Yes, this work clearly discussed the contributions that distinguish it from the previous ones.

**Summary And Contributions:**

This work constructed benchmark pairs for EOT (entropic optimal transport) and Schrödinger Bridge (SB), and tested existing neural EOT/SB solvers on the benchmark pairs. This work proposed a method to create pairs of probability distributions with analytically known EOT solutions and evaluate the approximated solutions based on the exact solutions. This work might provide a proper standard for evaluating and comparing the EOT/SB solvers,  which the community lacks now.

---

> ### Author Response · Authors · 2023-08-22
>
> Dear Reviewer p1pQ, Thank you for your comments. Here are the answers to your questions.
>
> **(1) The potential function is chosen to be a weighted log-sum-exp of quadratic functions. [...] However, this choice may conflict withgenerality and representativeness, which are important aspects, especially for a benchmark. [...] More consideration about generality is needed to further improve this work. Besides, it may be better to propose a dataset or a method to construct data that is more related to some real applications**.
>
> We acknowledge your concern about generalizability and provide additional experiments in the **general response to all the reviewers** (answer to question 1) and in **newly added Appendix H**. We empirically show that the class of benchmark distributions that can be constructed with the LSE potentials is broad. In particular, we even demonstrate that we can construct benchmark pairs which are similar to real-world single-cell biological datasets.
>
> **(2) The authors have tested existing neural EOT/SB solvers. However, the experimental results may not be convincing enough due to the lack of tuning hyperparameters (I think the authors are quite aware of this point).**
>
> We have taken into account your concerns about the dependence of the solvers on the hyper-parameters and addressed them in the **general response** (answer to question 2) and in the newly added **Appendix E**.
>
> **Concluding remarks**.
> We would be grateful if you could let us know if the explanations we gave have been satisfactory in addressing your concerns about our work. If so, we kindly ask that you consider increasing your score. We are also open to discussing any other questions you may have.

---

> > ### Author Response · Authors · 2023-08-27
> > **Looking forward to your final feedback**
> >
> > Dear Reviewer p1pQ,
> >
> > We thank you for your review and appreciate your time reviewing our paper.
> >
> > The end of the rebuttal phase is approaching. We would be grateful if we could hear your feedback regarding our answers to the reviews. We are happy to address any remaining points during the remaining period.
> >
> > Thanks in advance,
> >
> > the paper authors.

---

> ### Comment · Reviewer_p1pQ · 2023-08-28
>
> The authors' response has addressed my concerns.
>
> After careful consideration, I decided to raise my rating from 5 to 6.

---

> > ### Author Response · Authors · 2023-08-28
> > **An appreciation for considering the updated paper and rebuttal.**
> >
> > Dear reviewer p1pQ, we are glad that our answers have addressed your concerns. We thank you for the time you spent carefully considering our work and for raising your rating.

---

### Official Review · Reviewer_oGez · 2023-07-21
**Review of 'Building the Bridge of Schrödinger'**

**Rating:** 7
**Confidence:** 1
**Correctness:** The paper is correct, as far as I can…
**Clarity:** The paper is well-written.

**Strengths:**

The authors explain the limitations of existing methods and the knowledge gaps in the field. The interconnection of SB and EOT with diffusion models is articulated, and the potential application of these methods in generative modelling is discussed.

The theoretical grounding of the paper is strong, with detailed information on classic and weak OT formulations, the entropic OT formulation, and the interconnection between EOT and SB.

Despite some limitations (see below), the methodology provides a valuable tool for the evaluation of EOT/SB solvers and offers a detailed and carefully constructed approach to benchmarking in this complex and challenging field. The work fills a critical gap by introducing a first theoretically-grounded EOT/SB benchmark. This provides a standard way to test existing neural (continuous) solvers which previously did not exist.

**Additional Feedback:**

NA

**Documentation:**

Correct and provided in the paper.

**Limitations:**

See above

**Opportunities For Improvement:**

The paper might have benefited from further discussing the potential practical applications of this work or offering more specific examples of how this benchmarking tool could be used in the context of other machine-learning tasks.

The first aim proposes a generic method to create continuous pairs of probability distributions with an analytically known EOT solution. However, analytical solutions might not be available for all types of distributions or scenarios. In more complex or non-standard distributions, an analytically known solution might be impossible to derive, and hence, the pertinence of the benchmark is questionable.

The authors use log-sum-exp of quadratic functions to construct the pairs of distributions. I'm not an expert on the topic, and I wonder how this choice limits the generalizability of the benchmark and results.

Creating new benchmark pairs using the given methodology may require Monte Carlo Markov Chain (MCMC) to sample from them. This could introduce additional complexity and potential sources of error.

As the authors acknowledge themselves, most solvers’ performance significantly depends on the selected hyper-parameters that the authors do not optimize. Hence, the solvers' comparison provided in the paper is questionable.

**Relation To Prior Work:**

Described in the paper.

**Summary And Contributions:**

This paper contributes to the field of optimal transport (OT), Schrödinger Bridges (SB), and entropic optimal transport (EOT). The authors target a key challenge in the literature - the lack of non-trivial tests to evaluate methods' ability to solve SB or EOT problems - and propose a novel approach to tackle it. Namely, the authors propose a novel method for creating pairs of probability distributions with a known ground truth OT solution.

---

> ### Author Response · Authors · 2023-08-22
>
> Dear Reviewer oGez, thank you for your comments. Here are the answers to your questions.
>
> **(1) The paper might have benefited from further discussing the potential practical applications of this work or offering more specific examples of how this benchmarking tool could be used in the context of other machine-learning tasks.**
>
> As one of the promising applications of the EOT/SB methods is the analysis of single cell biological data, we have added an additional discussion on the creation of benchmark pairs similar to real single cell data [1, 2]. We have described this in the general response and in Appendix H of our paper. The important problem with single cell data is that there is no way to collect ground truth data about the map between two distributions $\mathbb{P}_0$ and $\mathbb{P}_1$. One of the possible ways to overcome this problem could be the use of constructed benchmark pairs similar to real data. Please also see the answer to your next question.
>
> **(2) [...] an analytically known solution might be impossible to derive, and hence, the pertinence of the benchmark is questionable. [...] The authors use log-sum-exp of quadratic functions to construct the pairs of distributions. [...] I wonder how this choice limits the generalizability of the benchmark and results.**
>
> We acknowledge your concern about generalizability and provide additional experiments in the **general response to all the reviewers** (answer to question 1) and in **newly added Appendix H**. We empirically show that the class of benchmark distributions that can be constructed with the LSE potentials is broad. In particular, we even demonstrate that we can construct benchmark pairs which are similar to real-world single-cell biological datasets.
>
> **(3) Creating new benchmark pairs using the given methodology may require Monte Carlo Markov Chain (MCMC) to sample from them. This could introduce additional complexity and potential sources of error.**
>
> We agree with your comment and would like to clarify one extra detail here. To create the images benchmark pairs and be able to sample from $\mathbb{P}\_0, \mathbb{P}\_{1}$, $\pi^{\star}(\cdot|x)$ and pairs from $\pi^{\star}$, we actually do not need MCMC. These benchmark pairs are constructed by using LSE potentials. This allows us to have closed-form conditional distributions $\pi^*(y|x)$, allowing all the above-mentioned sampling. However, to sample from the reverse conditional distributions $\pi^{*}(x|y)$ which are needed for **evaluation** of solvers but not for training, we indeed need MCMC. In the revised version, we have added a more detailed clarifications to the limitations (Section 6).
>
> **(4) As the authors acknowledge themselves, most solvers’ performance significantly depends on the selected hyper-parameters that the authors do not optimize. Hence, the solvers' comparison provided in the paper is questionable.**
>
> We have taken into account your concerns about the dependence of the solvers on the hyper-parameters and addressed them in the **general response** (answer to question 2) and in the newly added **Appendix E**.
>
> **Concluding remarks**.
> We would be grateful if you could let us know if the explanations we gave have been satisfactory in addressing your concerns about our work. If so, we kindly ask that you consider increasing your score. We are also open to discussing any other questions you may have.
>
> **Additional references.**
>
> [1] Koshizuka T., Sato I. Neural Lagrangian Schr\"{o} dinger Bridge: Diffusion Modeling for Population Dynamics //The Eleventh International Conference on Learning Representations. – 2022.
>
> [2] Bunne C. et al. Learning single-cell perturbation responses using neural optimal transport //bioRxiv. – 2021. – С. 2021.12. 15.472775.

---

> > ### Author Response · Authors · 2023-08-27
> > **Looking forward to your final feedback**
> >
> > Dear Reviewer oGez,
> >
> > We thank you for your review and appreciate your time reviewing our paper.
> >
> > The end of the rebuttal phase is approaching. We would be grateful if we could hear your feedback regarding our answers to the reviews. We are happy to address any remaining points during the remaining period.
> >
> > Thanks in advance,
> >
> > the paper authors.

---

> > > ### Comment · Reviewer_oGez · 2023-08-28
> > >
> > > I thank the authors for their detailed answers. The responses address my concerns and I appreciate the effort they have put into the rebuttal. I already gave a good score, which I maintain.

---

> > > > ### Author Response · Authors · 2023-08-28
> > > > **An appreciation for considering the updated paper and rebuttal.**
> > > >
> > > > Dear reviewer oGez, we sincerely appreciate the time and effort you have taken to review our response. We are pleased that our responses have addressed your concerns and that you have given our work a good rating.

---

### Official Review · Reviewer_UzdZ · 2023-07-25
**A method for producing dataset for SB Solvers**

**Rating:** 9
**Confidence:** 4
**Correctness:** Yes.
**Clarity:** Yes.

**Strengths:**

They has provided necessary theoretic result to pave the way for further study on the dataset for SB solvers. The dataset they obtained is of high quality.

**Additional Feedback:**

More experiments on the benchmark will be appreciated.

**Documentation:**

Yes, they provided a github link for this work.

**Ethics:**

No.

**Limitations:**

The  most solvers’ performance significantly depends on the selected hyper-parameters.
It may be a serious issue in adopting this method to generate dataset.

**Opportunities For Improvement:**

No obvious weakness.

**Relation To Prior Work:**

They have introduced the necessary background in a clear manner.

**Summary And Contributions:**

Nowadays, Diffusion process has been a prominent method in image-to-image translation tasks. To evaluate the solvers  of Schrodinger Bridge, this paper provides a method to producing such a dataset. They introduced and proved a set of theorems to justify their method.

---

> ### Author Response · Authors · 2023-08-22
>
> Dear Reviewer UzdZ, Thank you for your comments. Here are the answers to your questions.
>
> **(1) The most solvers’ performance significantly depends on the selected hyper-parameters. It may be a serious issue in adopting this method to generate dataset. [...] More experiments on the benchmark will be appreciated.**
>
> We have taken into account your concerns about the dependence of the solvers on the hyper-parameters and addressed them in the **general response** (answer to question 2) and in the newly added **Appendix E**.
>
> **Concluding remarks**.
> We would be grateful if you could let us know if the explanations we gave have been satisfactory in addressing your concerns about our work. We are also open to discussing any other questions you may have.

---

### Official Review · Reviewer_XL1t · 2023-07-25

**Rating:** 6
**Confidence:** 3

**Strengths:**

- Novel methodology: the work presents a novel, theoretically grounded, and rather generic approach to construct benchmark pairs of probability distributions with known EOT/SB solutions.

- Standardization: these benchmark problems fill a significant gap in the field and enable standardized testing of corresponding solvers.

- High-dimensional evaluation: the methodology enables the evaluation of solvers in high-dimensional spaces, such as image spaces, which are relevant to many generative modeling tasks.

**Additional Feedback:**

In Section 3, it would be great to give a high-level description of the construction (perhaps with a corresponding simple example).

**Clarity:**

The paper is generally well-written and structured. However, I think the bold typesetting should be reduced, and the language could be improved. Moreover, the equations run over the margin in Section A in the appendix.

**Correctness:**

The paper seems theoretically sounds and the rigor of the mathematical derivations is appropriate.

**Documentation:**

In the appendix and the corresponding GitHub repositories, sufficient detail is provided to support reproducibility.

**Limitations:**

Several limitations seem to be mentioned in the paper. Further limitations can also be found in "Opportunities For Improvement" above.
The section on societal impact could be improved, however, seems sufficient for the present paper.

**Opportunities For Improvement:**

- The use of quadratic LSE functions to construct distribution pairs may not fully represent practical scenarios, potentially limiting the generalizability of the benchmark.

- The use of $\mathrm{B}\mathbb{W}^2_2-\mathrm{UVP}$ for quantitative evaluation may not capture the full picture of the solvers' performance since only the first and second moments of the distributions are compared. The need to develop new evaluation metrics is acknowledged.

- While a reasonable number of solvers is presented and compared, it seems like significantly more hyperparameter tuning is needed to provide a fair comparison (since the default parameters are often far from being tuned for the considered examples). There is a corresponding disclaimer in the paper; however, this issue seems to limit the score of the experimental results severely.

- The high-dimensional image distributions seem to only be tractable by fitting a normalizing flow and performing MCMC in its latent space. Such a limitation should also be mentioned earlier in the paper.

**Relation To Prior Work:**

The paper outlines approaches to evaluate current solvers and also mentions relevant prior work in OT (e.g., [29,30]), but a more detailed comparison would be desirable (e.g., whether the benchmark results align with performance on downstream tasks or OT benchmarks for sufficiently small entropy regularization.).

**Summary And Contributions:**

The present paper proposes a generic methodology for evaluating continuous solvers for the Schrödinger Bridge (SB) problem and its equivalent, the Entropic Optimal Transport (EOT) problem. While these problems have recently gained interest in the field of generative modeling, the lack of standardized testing methods for neural EOT/SB solvers poses a challenge. Current benchmarks for neural Optimal Transport (OT) tasks are limited to specific formulations and do not generalize to the EOT problem. To address this gap, the authors propose a novel approach for generating pairs of probability distributions with analytically known EOT/SB solutions, which also works in high-dimensional spaces such as image spaces. The paper then uses this methodology to evaluate and compare common neural EOT/SB solvers.

---

> ### Author Response · Authors · 2023-08-22
> **Response (part I)**
>
> Dear Reviewer XL1t, thank you for your comments. Here are the answers to your questions.
>
> **(1) The use of quadratic LSE functions to construct distribution pairs may not fully represent practical scenarios, potentially limiting the generalizability of the benchmark.**
>
> We acknowledge your concern about generalizability and provide additional experiments in the **general response to all the reviewers** (answer to question 1) and in **newly added Appendix H**. We empirically show that the class of benchmark distributions that can be constructed with the LSE potentials is broad. In particular, we even demonstrate that we can construct benchmark pairs which are similar to real-world single-cell biological datasets.
>
> **(2) The use of BW-UVP for quantitative evaluation may not capture the full picture of the solvers' performance, since only the first and second moments of the distributions are compared. The need to develop new evaluation metrics is acknowledged.**
>
> We agree with the reviewer. At the same time, we would like to point out that even $B\mathbb{W}\_{2}^{2}$-UVP  allows to differentiate solvers by their ability to restore the target distribution and EOT plan. In fact, it is known [1] that the Bures-Wasserstein metric, i.e., $\mathbb{W}\_{2}$ between Gaussian approximations, provably **lower bounds** the actual Wasserstein-2:
> $$B\mathbb{W}\_{2}^{2}(\mathbb{P},\mathbb{Q})=\mathbb{W}\_{2}^{2}\big((\mathcal{N}(\mu\_{\mathbb{P}},\Sigma\_{\mathbb{P}}), \mathcal{N}(\mu\_{\mathbb{Q}},\Sigma\_{\mathbb{Q}})\big)\leq \mathbb{W}\_{2}^{2}(\mathbb{P},\mathbb{Q}).$$
> The latter is a powerful and explanatory metric, but is hard to compute from samples. Hence, $B\mathbb{W}\_{2}^{2}$ plays the role of its more computationally feasible replacement (one just needs to estimate means and covariances) which is still clearly related to $\mathbb{W}\_{2}^{2}$. For completeness, we also note that $B\mathbb{W}\_{2}^{2}$ is also known as the Freschet distance (FD) and is widely used as a part of the FID score to evaluate generative models [2]. We hope the above-mentioned arguments add even more validity to $B\mathbb{W}\_{2}^{2}$-UVP metric and provide extra intuition about why using it is reasonable.
>
> **(3) While a reasonable number of solvers is presented and compared, it seems like significantly more hyperparameter tuning is needed to provide a fair comparison (since the default parameters are often far from being tuned for the considered examples). There is a corresponding disclaimer in the paper; however, this issue seems to limit the score of the experimental results severely.**
>
> We have taken into account your concerns about the dependence of the solvers on the hyper-parameters and addressed them in the **general response** (answer to question 2) and in the newly added **Appendix E**.
>
> **(4) The high-dimensional image distributions seem to only be tractable by fitting a normalizing flow and performing
> MCMC in its latent space. Such a limitation should also be mentioned earlier in the paper.**
>
> Here we want to clarify the role of MCMC and normalizing flows that we used for the Celeba image benchmark pairs.
>
> We fit normalizing flows to model the Celeba image dataset and consider it as a distribution $\mathbb{P}_0$. Subsequently, by utilizing Log-Sum-Exponential (LSE) potentials, we create an alternative distribution, $\mathbb{P}_1$, along with its associated mapping $\pi^*(y|x)$. Importantly, we underscore that **our process of sampling** for solvers' training **does not need MCMC.**
>
> The utilization of LSE potentials, combined with the inherent low-dimensional nature of the image manifold, introduces Gaussian-like noise to the images present in the newly constructed distribution $\mathbb{P}_1$. Since the denoising problem is more interesting and similar to real applications, we consider the EOT problem from $\mathbb{P}_1$ to $\mathbb{P}_0$, i.e., from noised images to good images. We can still sample pairs $(x, y)\sim \pi^*(x, y)$ without using MCMC. We only need MCMC to construct the **test** data sets of the samples from $\pi^{\star}(x|y)\propto \pi^{\star}(y|x)\pi^{\star}(x)$, i.e., we want to recover several good images from one image with noise. We need to do this because we have proposed to use FID as the standard image metric. **So we only use MCMC to build the test data to compute the metric**, although we admit that using MCMC may introduce some bias into the FID metric. At the same time, the **normalizing flow is needed only to provide the density** $\pi^{*}(x)$ needed for MCMC in the Bayes formula above.
>
> To conclude, we want to say that MCMC and normalizing flows were used only for specific metric calculation which we picked and it is mainly question about the metrics rather than about benchmark pairs construction.

---

> > ### Author Response · Authors · 2023-08-22
> > **Response (part II)**
> >
> > **(5) Several limitations seem to be mentioned in the paper. Further limitations can also be found in "Opportunities For Improvement" above. The section on societal impact could be improved, however, seems sufficient for the present paper.**
> >
> > Following your suggestion, we have revised the paper and extended its limitations, see section 6.
> >
> > **Concluding remarks**.
> > We would be grateful if you could let us know if the explanations we gave have been satisfactory in addressing your concerns about our work. If so, we kindly ask that you consider increasing your score. We are also open to discussing any other questions you may have.
> >
> > **Additional references**.
> >
> > [1] Cuesta-Albertos, J. A., Matrán-Bea, C., & Tuero-Diaz, A. (1996). On lower bounds for the L 2-Wasserstein metric in a Hilbert space. Journal of Theoretical Probability, 9(2), 263-283.
> >
> > [2] Heusel, M., Ramsauer, H., Unterthiner, T., Nessler, B., & Hochreiter, S. (2017). Gans trained by a two time-scale update rule converge to a local nash equilibrium. Advances in neural information processing systems, 30.

---

> > > ### Author Response · Authors · 2023-08-27
> > > **Looking forward to your final feedback**
> > >
> > > Dear Reviewer XL1t,
> > >
> > > We thank you for your review and appreciate your time reviewing our paper.
> > >
> > > The end of the rebuttal phase is approaching. We would be grateful if we could hear your feedback regarding our answers to the reviews. We are happy to address any remaining points during the remaining period.
> > >
> > > Thanks in advance,
> > >
> > > the paper authors.

---

### Official Review · Reviewer_jvUZ · 2023-07-26
**A novel theoretical way of benchmark creation and sampling**

**Rating:** 6
**Confidence:** 4
**Correctness:** Yes.
**Clarity:** Yes.

**Strengths:**

This paper indeed provides clear guidelines for creating synthetic evaluation data which are theoretically sound. It will direct benefit the community of optimal transport study. And it will potentially benefit the research of generated model.

Many technical details introduced when creating the dataset could also be baseline methods that facilitate the following research.

**Additional Feedback:**

No

**Documentation:**

The code is in general easy to read and well structured. Readers could benefit more when there are high-level descriptions for each file or function.

**Limitations:**

Yes

**Opportunities For Improvement:**

This paper emphasizes more on "methods of creating datasets" rather than "benchmarking existing methods". To be an impactful benchmark, it should at least test a wide range of synthetic situations, or a few situations that are strongly related to the real-world data with supportive evidence. Therefore, this paper passes limited results for readers to choose better methods.

**Relation To Prior Work:**

Yes

**Summary And Contributions:**

This paper creates both synthetic benchmarks and real image-based benchmarks for both entropic optimal transport problems and Schrodinger bridge problems, where how well the methods predict optimal transport plan and optimal drift is evaluated. Notably, the suggested method for populating those benchmarks is theoretically sound and computationally efficient (by introducing log-sum-exp quadratic potentials to sample Gaussian mixtures). The methods are evaluated based on the Bures-Wasserstein metric and "conditional" Bures-Wasserstein metric The authors showcase how to populate the benchmarks with practical recipes. The related functionality is also provided in the code repository.

---

> ### Author Response · Authors · 2023-08-22
>
> Dear Reviewer jvUZ, thank you for your comments. Here are the answers to your question.
>
> **(1) This paper emphasizes more on "methods of creating datasets" rather than "benchmarking existing methods". To be an impactful benchmark, it should at least test a wide range of synthetic situations, or a few situations that are strongly related to the real-world data with supportive evidence. Therefore,  this paper passes limited results for readers to choose better methods.**
>
> Due to the fact that OT/SB theory is hard on its own, it is very natural that constructing benchmark pairs for OT/SB is a non-trivial task itself. In particular, prior to our work, the *only* EOT/SB scenario with known GT solutions was Gaussian-to-Gaussian mapping. Hence, we think this is reasonable that the main part of our paper emphasizes the theoretically-justified *methodology of creating the benchmark datasets (pairs)* rather than *benchmarking the solvers*.  One more reason for this is that without the appropriate datasets one is simply unable to evaluate the existing solvers.
>
> In fact, our main contribution is the benchmark construction methodology plus particular constructed benchmark pairs. The evaluation of existing EOT/SB solvers is provided in order to showcase the usefullness of our constructed benchmark pairs for providing the general understanding of solver's performance and their failure cases. The current field of EOT/SB lacks any comprehensive comparisons of existing methods (see Section 2 of our paper).
>
> At the same time, we understand, that it is indeed important to stock the scenarios (with known EOT/SB solutions) which are close to the real-world ones. In order to empower our benchmark with such practically-resembling use-cases we demonstrate that our proposed methodology for constructing benchmark pairs makes it possible to construct new examples of **benchmark pairs built on top of single-cell biological data**, see our General response, section **1(b)**. We emphasize that biological applications, in particular, single-cell data processing, is a promising direction for EOT/SB solvers, as validated by numerous researches [1,2,3,4,5,6,7,8]. Therefore, our newly proposed methodology may make it possible to compare existing and hypothetical methods on "close to reality" setups.
>
> **Concluding remarks**.
> We would be grateful if you could let us know if the explanations we gave have been satisfactory in addressing your concerns about our work. If so, we kindly ask that you consider increasing your score. We are also open to discussing any other questions you may have.
>
> [1] Schiebinger et. al. "Optimal-Transport Analysis of Single-Cell Gene Expression Identifies Developmental Trajectories in Reprogramming" // Cell, 2019.
>
> [2] Koshizuka T., Sato I. Neural Lagrangian Schr\"{o} dinger Bridge: Diffusion Modeling for Population Dynamics //The Eleventh International Conference on Learning Representations. – 2022.
>
> [3] Bunne C. et al. Learning single-cell perturbation responses using neural optimal transport //bioRxiv. – 2021. – С. 2021.12. 15.472775.
>
> [4] Tong, Alexander, et al. "Simulation-free Schr\" odinger bridges via score and flow matching." arXiv preprint arXiv:2307.03672 (2023).
>
> [5] Bunne C. et al. Proximal optimal transport modeling of population dynamics //International Conference on Artificial Intelligence and Statistics. – PMLR, 2022. – С. 6511-6528.
>
> [6] Moon, Kevin R., et al. "Visualizing structure and transitions in high-dimensional biological data." Nature biotechnology 37.12 (2019): 1482-1492.
>
> [7] Tong A. et al. Trajectorynet: A dynamic optimal transport network for modeling cellular dynamics //International conference on machine learning. – PMLR, 2020. – С. 9526-9536.
>
> [8] Huizing G.-J. et. al., "Optimal transport improves cell–cell similarity inference in single-cell omics data" //Bioinformatics, 2022.

---

> > ### Author Response · Authors · 2023-08-27
> > **Looking forward to your final feedback**
> >
> > Dear Reviewer jvUZ,
> >
> > We thank you for your review and appreciate your time reviewing our paper.
> >
> > The end of the rebuttal phase is approaching. We would be grateful if we could hear your feedback regarding our answers to the reviews. We are happy to address any remaining points during the remaining period.
> >
> > Thanks in advance,
> >
> > the paper authors.

---

### Author Response · Authors · 2023-08-22
**General Response (part I)**

Dear reviewers, thank you for taking the time to review our paper. We are pleased that you find our approach to be a novel methodology with a strong theoretical basis for generating data for evaluating EOT/SB methods (jvUZ, XL1t, oGez, UzdZ), and that the code implementation is well organized and easy to follow (p1pQ). We are pleased that you find our approach to construct pairs of distributions with known EOT/SB solutions to be an important and valuable tool that will benefit the community (p1pQ) and help in the evaluation and further development of new methods for solving EOT/SB problems (oGez, XL1t, jvUZ).

We uploaded the **revised version** of our paper and its appendix (changes are highlighted with the **blue color**) and added extra code examples to our repository. The main changes are:

- New **Appendix H** and folder *benchmark\_construction\_examples* in repository with examples of constructing benchmark pairs from data (for reviewers jvUZ, XL1t, oGez, p1pQ).

- New **Appendix E** containing the hyperparameters study of EOT/SB solvers (for reviewers XL1t, UzdZ, oGez, p1pQ).

We discuss these changes throughout the answers to your common questions below.

**1. Expressiveness of LSE potentials; constructing benchmark pairs that are similar to real data (for reviewers jvUZ, XL1t, oGez, p1pQ).**

We see that one of the main concerns is whether the usage of LSE potentials allows us to generate pairs of distributions that are complex enough and similar to real data. We provide a two-fold answer here.

**(a) Theoretical significance of our methodology; capabilities of approximating complex pairs.**

We would like to emphasize that our key theoretical result (Theorem 3.2 of Section 3.2) allows constructing pairs of **arbitrary complexity** (by picking any complex potential $f^{*}$) but may require MCMC to sample from them. At the same time,the class of existing continuous distributions with the analytically known EOT/SB solution is narrow (lines 30-31) as it consists only of plans between two multivariate Gaussians. Taking this into account, we believe that *our approach is already a step forward to a more general class of EOT/SB solutions.*

In our practical construction, we employ log-sum-exp potentials $f^{*}$ exclusively because they allow to avoid using MCMC (Propositions 3.3, 3.4). These potentials yield a kind of Gaussian mixture-like approximation of SB/EOT. Due to this, we think that it may satisfy the *universal approximation* property just like the basic Gaussian mixtures which are widely used for the density estimation task [9]. Although we do not provide a theoretical result here, our next part of the answer below shows that empirically LSE potentials allow approximating rather complex real data distributions.

**(b) Practical illustrations: new examples of complex benchmark pairs constructed from real data.**

To further support that the proposed class of EOT/SB solutions given by LSE potentials is rather large, we provide examples of constructing benchmark pairs that closely resemble real data. Recent articles have emphasized the fruitful branch of applications of EOT/SB in the analysis of single-cell biological data [1, 2, 3, 4, 5, 7]. Below we will show that LSE potentials can be used to construct complex distribution pairs, **including those that closely resemble real-world single cell data [1, 2, 6].** To construct benchmark distribution pairs resembling given data, we utilize original and target datasets from ground truth distributions $\mathbb{P}_0$ and $\mathbb{P}_1$, respectively.

**Our recipe is as follows.** We simply initialize the LSE potential $f^*(y) = \epsilon \log \sum_{n=1}^N w_n \mathcal{Q}(y|b_n, \epsilon^{-1}A_n)$ with cluster centers $b_n$ from K-means on the target data from $\mathbb{P}\_{1}$ and set $A_n = \lambda I$ (where $\lambda$ is chosen manually), $w_{n}=\frac{1}{N}$. We empirically found that this heuristical initialization strategy already leads to a distribution
$ d\widehat{\mathbb{P}}\_1(y) = d\pi^*_1(y)=\int d\pi^*(y|x) d\mathbb{P}\_0(x) ,$

that is similar to a Gaussian mixture approximation of the actual target data $\mathbb{P}_{1}$ as soon as one manages to find appropriate $\lambda$. Recall that here $\frac{d\pi^{\star}(y|x)}{dy}\propto \exp\frac{f^{\star}(y)-\frac{1}{2}\|x-y\|^{2}}{\epsilon}$ by our benchmark construction.

In the **newly added Appendix H**, we provide several examples of the construction of benchmark pairs from data and, in particular, show that parameter $\lambda$ can be easily chosen manually. We consider several 2D setups and two single cell setups mentioned in previous works [1, 2].  For all these setups, we add **three ipython notebooks** (in folder *benchmark_construction_examples* of the GitHub repository) that demonstrate the creation of the benchmark pairs for each considered setup.

---

> ### Author Response · Authors · 2023-08-22
> **General response (part II)**
>
> To qualitatively check that our built target distribution $\widehat{\mathbb{P}}_1$ is indeed similar to the ground truth target distribution $\mathbb{P}_1$, we fit the TSNE projection to a combined dataset of real and generated samples for single-cell setups [1, 2] and then compare their projections and present results in the Figures of Appendix H. For data from [2], we additionally use the same quantitative metrics as [2] to evaluate how well constructed plan maps original dataset to the target dataset (on the test data). Surprisingly, we find that our strategy performs even better than several previously proposed methods developed specifically for this biological task.
>
> **2. Hyperparameter tuning (for reviewers XL1t, UzdZ, oGez, p1pQ).**
>
> The second main concern of our paper is whether the hyperparameters used for the considered methods are adequate for method comparison. Here we would like to remind the reviewers that we take the code of each method from its official repository and use the same or similar hyperparameters that have already been used by the authors for similar problems. *Selecting the best hyperparameters is very challenging due to the fact that each method has its own underlying principles and hyperparameters. Overall, selection of the best hyperparameters is not the main goal of our study, which focuses on the generic methodology to construct benchmark pairs plus particular constructed pairs.*
>
> To address the reviewers' comment, for each of the methods considered, we have selected several **key hyperparameters** and carried out supplementary evaluation of their effect on the model quality. For this, we used our mixtures benchmark pairs in dimension $D=64$ and $\epsilon=1$. These additional experiments aimed to quantify the methods' sensitivity to hyperparameter variations and can be found in **newly added Appendix E**.
>
> We find that the solvers show a relatively weak dependence on the hyperparameters around those used for comparison, and generally require significantly more training iterations or number of samples used to achieve slightly better results.
>
> **3. Additional comparison with MLE-SB using neural network (extra comment for all the reviewers).**
>
> The authors of the MLE-SB solver [8] use Gaussian processes in their paper, whereas all other SB methods rely on neural networks. After submitting our paper to the current conference, we submitted it to arxiv as well. The authors of MLE-SB had the opportunity to read our preprint published on arXiv. They contacted us via an e-mail and recommended us incorporating neural networks in their method for a more fair comparison. In response, we replaced the Gaussian process approximator in the MLE-SB solver with a neural network. The outcomes of this adjustment are presented in Table 2 of the revised paper and Tables 4 and 5 in Appendix B of the revised supplementary material. Notably, employing neural networks in MLE-SB yields favorable results, comparable to those achieved by other SB solvers.
>
> **Concluding remarks.** We would be grateful if you could let us know if the explanations we gave have been satisfactory in addressing your concerns about our work. If so, we kindly ask that you consider increasing your score. We are also open to discussing any other questions you may have.
>
> **Additional references.**
>
> [1] Koshizuka T., Sato I. Neural Lagrangian Schr\"{o} dinger Bridge: Diffusion Modeling for Population Dynamics //The Eleventh International Conference on Learning Representations. – 2022.
>
> [2] Bunne C. et al. Learning single-cell perturbation responses using neural optimal transport //bioRxiv. – 2021. – С. 2021.12. 15.472775.
>
> [3] Bunne C. et al. The Schrödinger Bridge between Gaussian Measures has a Closed Form //International Conference on Artificial Intelligence and Statistics. – PMLR, 2023. – С. 5802-5833.
>
> [4] Tong, Alexander, et al. "Simulation-free Schr\" odinger bridges via score and flow matching." arXiv preprint arXiv:2307.03672 (2023).
>
> [5] Bunne C. et al. Proximal optimal transport modeling of population dynamics //International Conference on Artificial Intelligence and Statistics. – PMLR, 2022. – С. 6511-6528.
>
> [6] Moon, Kevin R., et al. "Visualizing structure and transitions in high-dimensional biological data." Nature biotechnology 37.12 (2019): 1482-1492.
>
> [7] Tong A. et al. Trajectorynet: A dynamic optimal transport network for modeling cellular dynamics //International conference on machine learning. – PMLR, 2020. – С. 9526-9536.
>
> [8] Vargas F. et al. Solving schrödinger bridges via maximum likelihood //Entropy. – 2021. – Т. 23. – №. 9. – С. 1134.
>
> [9] Nguyen, T. T., Nguyen, H. D., Chamroukhi, F., & McLachlan, G. J. (2020). Approximation by finite mixtures of continuous density functions that vanish at infinity. Cogent Mathematics & Statistics, 7(1), 1750861.

---

### Decision · Program_Chairs · 2023-09-22

**Decision:**

Accept (Poster)

**Comment:**

This work presents an interesting benchmark for optimal transport problems, which have become increasingly popular in the machine learning community. In particular, the authors focus on entropic optimal transport and Schrodinger bridge, and provide benchmarks based on both synthetic and real imaging data. While reviews ranged from marginal to strong support for the paper, they are all above acceptance threshold, and I believe it is generally agreed the paper would be of sufficient interest and significance for this track, and therefore I recommend acceptance.